# Prioritization of candidate causal genes for asthma in susceptibility loci derived from UK Biobank

Kim Valette[1], Zhonglin Li[1], Valentin Bon-Baret[1], Arnaud Chignon[1], Jean-Christophe Bérubé[1], Aida Eslami[1], Jennifer Lamothe[1], Nathalie Gaudreault[1], Philippe Joubert[1], Ma'en Obeidat[2], Maarten van den Berge[3], Wim Timens[4], Don D. Sin[2], David C. Nickle[5], Ke Hao[6], Catherine Labbé[1], Krystelle Godbout[1], Andréanne Côté[1], Michel Laviolette[1], Louis-Philippe Boulet[1], Patrick Mathieu[1], Sébastien Thériault[1,7] & Yohan Bossé[1,8 ✉]

To identify candidate causal genes of asthma, we performed a genome-wide association study (GWAS) in UK Biobank on a broad asthma definition (n = 56,167 asthma cases and 352,255 controls). We then carried out functional mapping through transcriptome-wide association studies (TWAS) and Mendelian randomization in lung (n = 1,038) and blood (n = 31,684) tissues. The GWAS reveals 72 asthma-associated loci from 116 independent significant variants ($P_{GWAS}$ < 5.0E-8). The most significant lung TWAS gene on 17q12-q21 is *GSDMB* ($P_{TWAS}$ = 1.42E-54). Other TWAS genes include *TSLP* on 5q22, *RERE* on 1p36, *CLEC16A* on 16p13, and *IL4R* on 16p12, which all replicated in GTEx lung (n = 515). We demonstrate that the largest fold enrichment of regulatory and functional annotations among asthma-associated variants is in the blood. We map 485 blood eQTL-regulated genes associated with asthma and 50 of them are causal by Mendelian randomization. Prioritization of druggable genes reveals known (*IL4R*, *TSLP*, *IL6*, *TNFSF4*) and potentially new therapeutic targets for asthma.

[1] Institut universitaire de cardiologie et de pneumologie de Québec, Université Laval, Quebec, QC, Canada. [2] The University of British Columbia Centre for Heart Lung Innovation, St Paul's Hospital, Vancouver, BC, Canada. [3] Department of Pulmonology, University of Groningen, University Medical Center Groningen, GRIAC Research Institute, Groningen, The Netherlands. [4] Department of Pathology and Medical Biology, University of Groningen, University Medical Center Groningen, GRIAC Research Institute, Groningen, The Netherlands. [5] Gossamer Bio, San Diego, CA, USA. [6] Department of Genetics and Genomic Sciences, Icahn School of Medicine at Mount Sinai, New York, NY, USA. [7] Department of Molecular Biology, Medical Biochemistry and Pathology, Laval University, Quebec, QC, Canada. [8] Department of Molecular Medicine, Laval University, Quebec, QC, Canada. ✉email: yohan.bosse@criucpq.ulaval.ca

Asthma is still causing 420,000 deaths per year and afflicts 300 million individuals worldwide[1]. Our understanding of the genetics of asthma has progressed following the completion of large GWAS by international consortia, namely GABRIEL[2], EVE[3], Trans-National Asthma Genetic Consortium[4], and Consortium on Asthma among African Ancestry Populations[5]. More recently, two groups of investigators tapped into the UK Biobank resource to delineate the genetics of childhood-onset vs. adult-onset asthma[6,7]. Together, approximately 200 genetic loci have been associated with asthma through GWAS. One remaining challenge arising from these GWAS results is to find the underlying causal genes.

In parallel to GWAS, large expression quantitative trait loci (eQTL) datasets have been generated in asthma-relevant tissues, such as the lung and blood[8–10]. By leveraging these eQTL datasets, previous studies have identified genes whose expression levels were associated with asthma genetic variants[11–13]. With evolving bioinformatics approaches, GWAS and eQTL results can be integrated at the genome-wide scale to (1) find shared association signals using colocalization[14], (2) identify genes whose genetically-predicted gene expression levels are associated with asthma using a transcriptome-wide association study (TWAS)[15], and (3) infer causal association between genetically-determined gene expression and asthma using Mendelian randomization. Here, we hypothesized that existing omics datasets coupled with new bioinformatics tools will prioritize candidate causal genes underlying asthma susceptibility loci revealed by GWAS.

The objective of this study was to identify candidate causal genes of asthma in lung and blood tissues. This was achieved in two steps. First, performing a case–control GWAS on a broad asthma definition in UK Biobank in order to physically define chromosome regions associated with asthma. UK Biobank was selected as it is the largest case–control series of asthma available. Second, prioritizing candidate genes by mapping the effects of asthma-associated variants on protein-coding genes, gene expression, and chromatin interaction sites using multiple approaches such as TWAS, colocalization, and Mendelian randomization. Briefly, we describe 72 physically-defined asthma susceptibility loci in UK Biobank, identify 55 significant lung TWAS genes as well as 50 blood genes causally associated with asthma by Mendelian randomization, and finally prioritize 40 druggable genes as therapeutic targets for asthma.

## Results

**Asthma GWAS in UK Biobank**. In total, 56,167 asthma cases and 352,255 controls of White British ancestry were selected from UK Biobank (see "Methods" section). Demographics and clinical characteristics of cases and controls are in Table 1. The number of cases corresponds to an asthma population prevalence of 13.8%, which is consistent with the UK lifetime prevalence of patient-reported clinician-diagnosed asthma of 15.6%[16]. The granularity of asthma cases defined based on self-reported questionnaires, hospital records (ICD-9 and ICD-10), and primary care records is provided in Supplementary Fig. 1. For GWAS analysis, 35,270,583 single nucleotide polymorphisms (SNPs) (filtered by minor allele frequency >0.0001 and imputation info score >0.3) were available for genetic association testing following standard quality controls and imputation. We observed no evidence of inflation in the test statistics with $\lambda = 1.029$ (Supplementary Fig. 2). The SNP-heritability on the liability scale was estimated at 11.3%. In total, 14,742 SNPs reached genome-wide significance ($P_{GWAS} < 5.0E-8$) at 73 physically defined loci. Figure 1 shows the Manhattan plot and individual loci are listed in Supplementary Data 1. Seven of these loci are novel, with no genetic variant associated with asthma in the literature published before January 1st, 2020. The locus 7p14 is characterized by only one rare SNP that passed the significance threshold (rs576468798, $P_{GWAS} = 2.00E-8$, imputation info = 0.61, Supplementary Fig. 3). Allele frequencies in asthma cases (0.00033) and controls (0.00013) range within those observed in reference populations (TOPMed = 0.00015, 1000G European = 0.0006). Nevertheless, we discarded this locus as more validation is needed to robustly establish its association with asthma. Regional plots for the 6 remaining loci are provided in Fig. 2. We checked for potential replication for the novel loci in summary statistics from the Trans-National Asthma Genetic Consortium comparing 19,954 European ancestry cases and 107,715 European ancestry controls[4]. The sentinel variants or the next most significant variants overlapping with the Trans-National Asthma Genetic Consortium were not associated with asthma ($P_{GWAS} > 0.05$). We also evaluated the number of independent association signals within the 72 loci by conditional analysis. Sixteen loci had more than one independent association signals ranging from 2 to 9 independent signals by locus, except for the MHC locus, where we observed 12 independent signals. In total, 116 independent associations with asthma risk at a $P_{GWAS} < 5.0E-8$ were observed (Supplementary Data 2). We report four novel signals, two independent signals at the MHC locus (rs2517761 and rs2523430) and two at the 1q21-FLG locus (rs185433896 and rs558312428), that were independent ($r^2 < 0.1$) from asthma-associated variants reported in the literature (Supplementary Data 3). Genetic association results for previous asthma GWAS

---

**Table 1 Demographics and clinical characteristics of asthma cases and controls in the UK Biobank.**

|  | Case $n = 56,167$ | Control $n = 352,255$ |
|---|---|---|
| Sex (% male) | 42.5 | 46.4 |
| Age (mean and range) | 56.5 (40–71) | 57.0 (39–73) |
| BMI (kg/m²) (mean and range) | 28.2 (13.1–69.0) [212] | 27.3 (12.1–74.7) [1079] |
| Smoking status (%) | [247] | [247] |
| Never smokers | 53.2 | 54.6 |
| Former smokers | 36.3 | 35.0 |
| Current smokers | 10.1 | 10.1 |
| Lung function (mean and range) |  |  |
| FEV1 (L) | 2.71 (2.08–5.89) [8705] | 3.08 (2.63–5.99) [32,443] |
| FVC (L) | 3.75 (3.04–7.95) [8688] | 3.96 (3.23–7.99) [32,383] |
| FEV1 (L)/FVC (L) | 0.72 (0.24–1) | 0.78 (0.17–1) |
| PEF (L/min) | 431 (380–995) [8068] | 440 (361–999) [28,123] |
| Atopy (%) | 45 [59] | 21 [387] |
| Eosinophil count (g/L) (mean and range) | 0.22 (0–5.4) [1780] | 0.17 (0–9.6) [11,039] |

Number of missing values is shown in square brackets when applicable.
*BMI* body mass index, *FEV1* forced expiratory volume in 1s, *FVC* forced vital capacity, *PEF* peak expiratory flow.

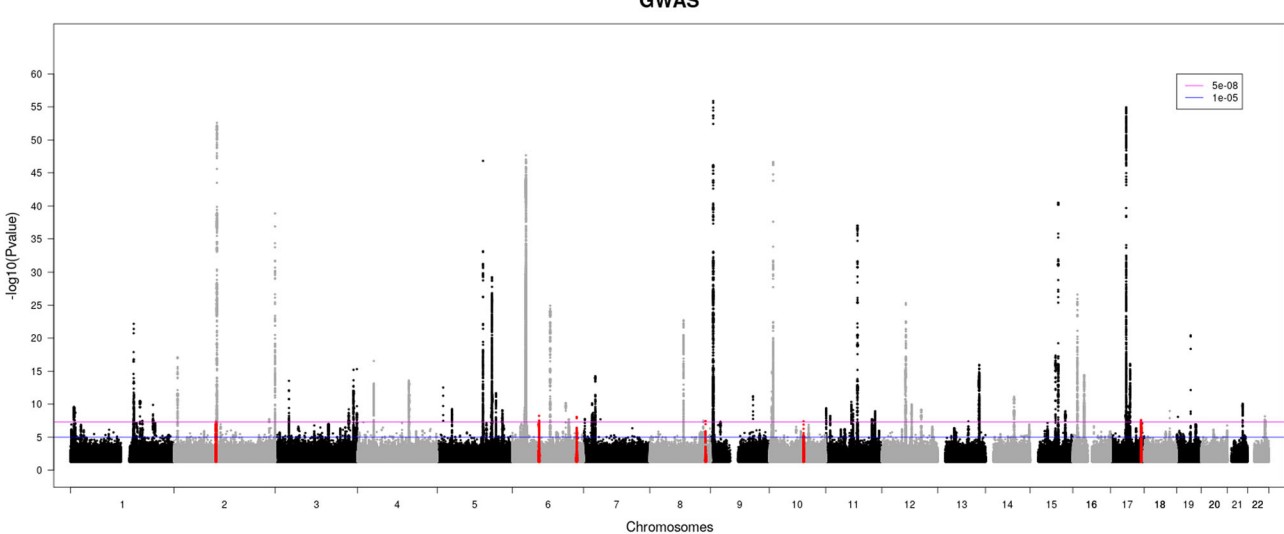

**Fig. 1 Manhattan plot of the GWAS on asthma in UK Biobank.** The GWAS was performed in 56,167 asthma cases and 352,255 controls. The $y$ axis represents $P$ value in −log10 scale. The horizontal blue and magenta lines indicate $P$ value of $1 \times 10^{-5}$ and $5 \times 10^{-8}$, respectively. Novel asthma loci are in red. Genetic variants with $P$ value > 0.05 were removed to limit the digital information of the Figure.

signals that are not significant in this study are provided in Supplementary Data 4.

**GWAS sensitivity analysis.** Three alternative study designs (2, 3, and 4) were evaluated to investigate the potential confounding effects of other lung diseases, smoking and allergy. The aforementioned results are considered study design 1 and the main analysis. This design was selected to maximize sample size and statistical power. Study design 2 excluded cases and controls with chronic obstructive pulmonary disease, emphysema, chronic bronchitis, interstitial lung disease, or alpha-1 antitrypsin deficiency ($n =$ 47,391 cases and 340,033 controls), because similarities in their clinical presentation can result in misclassification of cases and controls. Study design 3 excluded cases and controls with a positive smoking history ($n =$ 21,097 cases and 136,586 controls). This analysis was done to further evaluate the potential confounding effects of smoking-related lung disease, most particularly chronic obstructive pulmonary disease, on the asthma case–control status. Study design 4 excluded controls with atopy including hay fever, allergic rhinitis, and eczema/atopic dermatitis ($n =$ 56,167 cases and 268,142 controls). This study design explored the impact of excluding from the control group individuals who suffer from other genetically correlated allergic diseases, which may help to delineate unique vs. shared genetic etiology of asthma and allergy. Case–control genetic association analyses were thus performed on these three alternative study designs. Supplementary Fig. 4 compared the effect size estimates of the 72 sentinel asthma-associated variants discovered in study design 1 with the other study designs. Overall, the effect size estimates were highly similar. The single and most extreme discrepancy was observed for SNP rs558269137 (causing a frameshift in the filaggrin gene p.Ser761CysfsX36) at 1q21.3 with odds ratios (ORs) of 1.33 (95% CI, 1.26–1.41) in study design 1, 1.25 (1.14–1.37) in study design 2, 1.18 (1.02–1.36) in study design 3, and 1.27 (1.16–1.39) in study design 4. The results for the sentinel variant at each asthma locus and for the four study designs are provided in Supplementary Data 5.

**Functional annotation of coding SNPs.** Our first strategy to prioritize target genes within GWAS-nominated asthma loci was

to identify deleterious coding variants. This step was performed in FUMA[17]. FUMA takes GWAS summary statistics and derives its own list of independent significant variants ($P_{GWAS} < 5.0E-8$ and $r^2 < 0.6$ based on 1000G EUR). All known proxies that have $r^2 \geq 0.6$ with one of the independent significant variants, regardless of being available in the GWAS input file, are then considered candidate variants and included for further annotation. Supplementary Data 6 shows 354 exonic candidate variants identified by this method. These variants are located in 27 loci and two-thirds (236 out of 354) of these variants were located in the MHC locus. The most significant deleterious variants at this locus were rs9269958 in *HLA-DRB1* and rs2855430 in *COL11A2* with CADD scores of 57 and 33, respectively. However, association signals for these variants ($P_{GWAS} = 6.43E-5$ and 6.83E −7) were much smaller compared to the sentinel variant (rs9273386, $P_{GWAS} = 2.11E-48$). The extent of LD at this locus precluded firm conclusion. Outside of the MHC locus, we identified eight nonsynonymous variants and 1 stop-gain variant with CADD score >20 located at 7 loci (Table 2). Genes of known biological relevance were identified including filaggrin (*FLG*) on 1q21 and toll like receptor 10 (*TLR10*) on 4p14. On 17q12-q21, three potential target genes were identified, namely, *ERBB2*, *STARD3*, and *GSDMA*. In terms of effect sizes, the absolute beta values for the nine variants listed in Table 2 range from 0.03 to 0.22 (corresponding to ORs from 1.03 to 1.25). The largest effect size was observed for variant rs61816761 causing a G to A substitution (c.16819G>A) that occurs in exon 3 of the *FLG* gene, resulting in a stop instead of an arginine in codon 501 (p. Arg501Ter). However, the effect sizes of these coding variants were within the range observed for the 72 sentinel asthma-associated variants with ORs from 1.03 to 1.33. Noticeably, the largest effect size among sentinel variant was also observed for an independent deleterious coding variant in the *FLG* gene (rs558269137, p.Ser761CysfsX36). All genes in Table 2 have been reported in previous asthma GWAS. Overall, the yield of candidate genes by mapping of deleterious coding variants was relatively low. This is consistent with previous GWAS results on asthma that showed more genetic associations in noncoding regions of the genome, and suggests that most of the risk loci are likely to act through gene regulation.

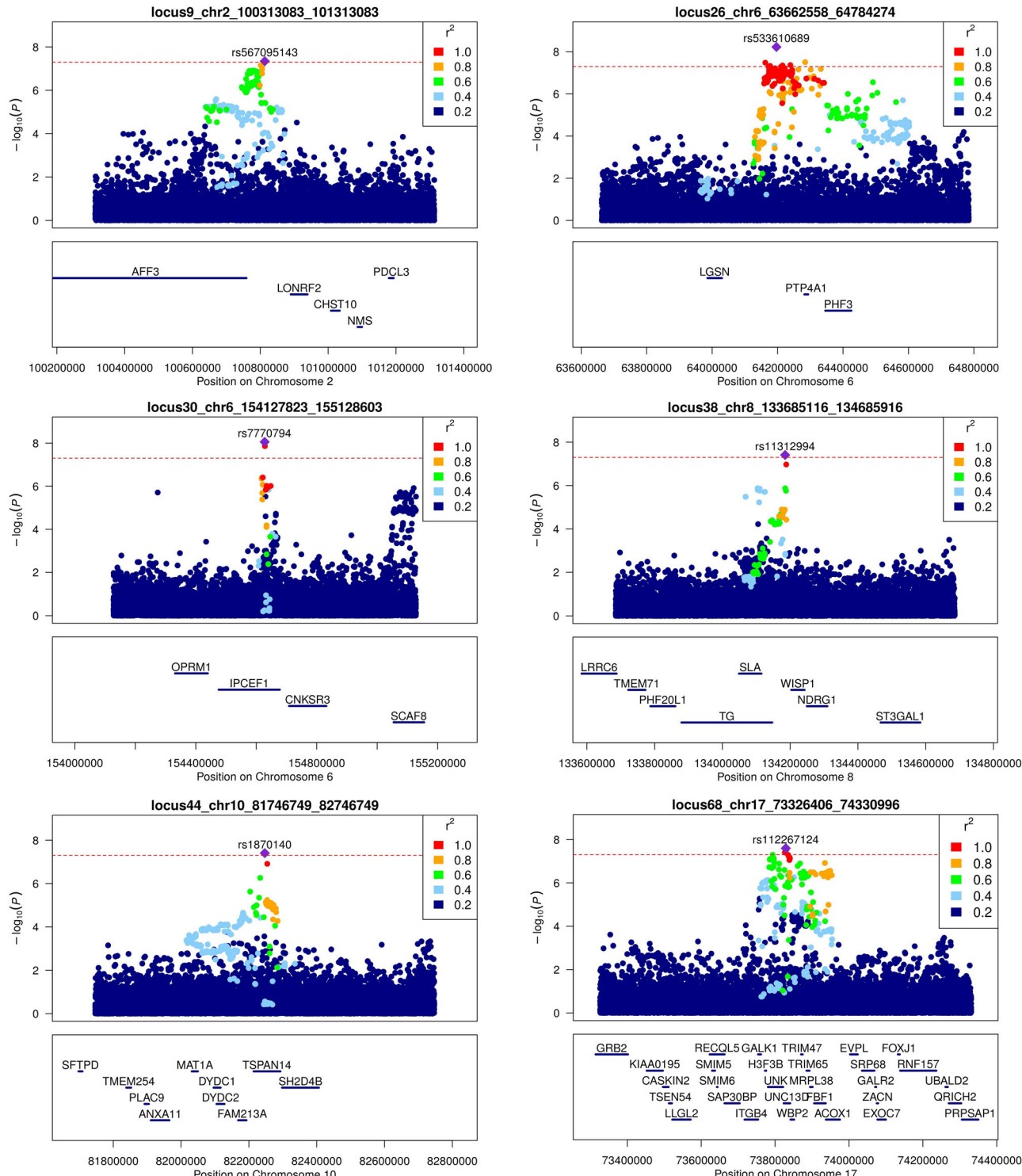

**Fig. 2 Regional plots showing the six new asthma-associated loci.** The y axis shows the P value in −log10 scale for SNPs upstream and downstream of the sentinel SNP (purple diamond). The extent of linkage disequilibrium (LD; $r^2$ values) for all SNPs with the sentinel SNP is indicated by colors. The location of genes is shown at the bottom. SNPs are plotted based on their chromosomal position on build 37.

**Asthma TWAS in lung tissue**. Summary statistics from the UK Biobank GWAS were integrated with our lung eQTL dataset[8] ($n = 1038$) to perform a TWAS on asthma. The full summary statistics for the asthma TWAS in lung tissue are available in the Supplementary Data 21. A total of 55 gene-asthma associations (corresponding to 69 probe sets) reached genome-wide significance ($P_{TWAS} < 2.51E{-}6$) (Fig. 3 and Supplementary Data 7).

Among the 55 lung TWAS genes, nine are novel for asthma. This includes *CSF2* and *HSPA4* on 5q31, *TRIM10* on 6p22-p21, *C9orf38* on 9p24, *TSPAN14* on 10q23, *FAM62A* on 12q13, *MAP2K5* on 15q22-q23, *CRKRS* and *PERLD1* on 17q12-q21.2. The lung TWAS genes associated with asthma were enriched in the Kyoto Encyclopedia of Genes and Genomes (KEGG) for asthma ($P_{adjusted} = 0.003$), antigen processing and presentation

**Table 2 Deleterious coding SNPs associated with asthma or in LD with asthma-associated SNPs outside of the MHC locus.**

| Chr | Chr band | rsID | PositionGRCh37 | $P_{GWAS}$ | Beta | CADD | Gene symbol | Gene name |
|-----|----------|------|----------------|-----------|------|------|-------------|-----------|
| 1 | 1p36 | rs2230624 | 12,175,658 | 1.99E−9 | −0.16 | 22.1 | *TNFRSF8* | TNF receptor superfamily member 8 |
| 1 | 1q21 | rs61816761 | 152,285,861 | 3.95E−22 | 0.22 | 36 | *FLG-AS1/ FLG* | Filaggrin |
| 4 | 4p14 | rs11096957 | 38,776,491 | 2.49E−10 | −0.05 | 21.9 | *TLR10* | Toll like receptor 10 |
| 5 | 5p15 | rs16903574 | 14,610,309 | 5.3E−12 | 0.09 | 22.6 | *FAM105A/ OTULINL* | OTU deubiquitinase with linear linkage specificity like |
| 11 | 11q13 | rs12146493 | 65,547,333 | 7.69E−6 | −0.03 | 22.2 | *AP5B1* | Adapter related protein complex 5 subunit beta 1 |
| 12 | 12q21 | rs3763978 | 71,533,534 | 2.6E−10 | −0.04 | 24.5 | *TSPAN8* | Tetraspanin 8 |
| 17 | 17q12 | rs1058808 | 37,884,037 | 1.94E−26 | −0.07 | 23.5 | *ERBB2* | erb-b2 receptor tyrosine kinase 2 |
| 17 | 17q12 | rs1877031 | 37,814,080 | 4.71E−22 | −0.07 | 23.1 | *STARD3* | StAR related lipid transfer domain containing 3 |
| 17 | 17q21 | rs3894194 | 38,121,993 | 7.95E−33 | 0.08 | 21.9 | *GSDMA* | Gasdermin A |

All variants are nonsynonymous except rs61816761 in the filaggrin gene that is a stop-gain.

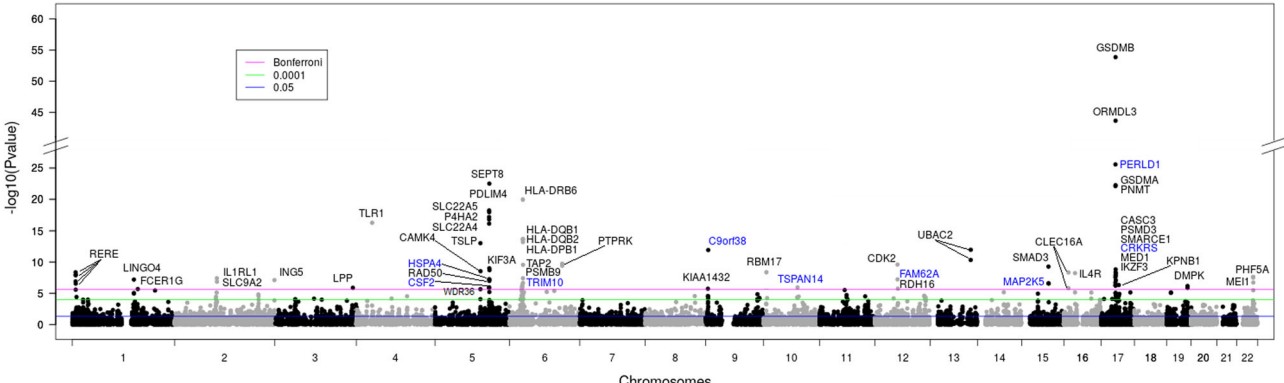

**Fig. 3 Manhattan plot of the TWAS on asthma integrating the UK Biobank GWAS and the lung eQTL dataset.** Each dot represents the association between predicted gene expression and asthma for a specific probe/transcript. P values for gene expression-asthma associations are on the y axis in −log10 scale. The blue, green, and magenta horizontal lines represent $P_{TWAS}$ of 0.05, 0.0001, and 2.51E−6 (Bonferroni), respectively. Annotations for genome-wide significant probes/transcripts that passed Bonferroni correction are indicated. Genes in blue have not been reported in previous asthma GWAS.

($P_{adjusted}$ = 0.003) and Th17 cell differentiation ($P_{adjusted}$ = 0.004) (Supplementary Data 8). Fifty-three of these genes are located in 21 distinct asthma-associated loci identified in the UK Biobank GWAS (Table 3). Supplementary Fig. 5 shows the most significant lung TWAS genes per asthma-associated loci. The most significant TWAS signal is at the well-known asthma-associated locus on chromosome 17q12-q21. The lead TWAS target gene at this locus is *GSDMB* ($P_{TWAS}$ = 1.42E−54). However, nine additional statistically significant TWAS genes are identified including *ORMDL3* ($P_{TWAS}$ = 2.12E−44), *GSDMA* ($P_{TWAS}$ = 5.52E−23), and *PNMT* ($P_{TWAS}$ = 7.87E−23). LocusCompare plots showing the colocalization events for these TWAS genes on 17q12-q21 are provided in Supplementary Fig. 6, and show that the *P* value distribution of eQTL for *GSDMB* colocalized better with that of the GWAS. The direction of effects, i.e., whether lower or higher predicted expression of these genes increased asthma risk are presented in Table 3, along with other TWAS genes found at asthma-associated loci. TWAS genes of known biological interest in asthma include *IL1RL1* on 2q12, *TLR1* on 4p14, *TSLP* on 5q22, *SMAD3* on 15q22-q23, and *IL4R* on 16p12.

TWAS can also reveal novel risk loci owing to the resulting power of combining GWAS and eQTL. In this study, two TWAS genes are located in genomic loci that did not reach statistical significance in the GWAS. This includes the gene encoding the gamma chain of the high-affinity IgE receptor (*FCER1G*, z = 4.74, $P_{TWAS}$ = 2.13E−6) on chromosome 1q23.3 playing a key role in

allergic reactions and DM1 protein kinase (*DMPK*, z = 4.83, $P_{TWAS}$ = 1.37E−6) on chromosome 19q13.32 with cellular antioxidant and pro-survival properties[18].

GTEx lung was used to validate the TWAS results. For the two novel asthma risk loci, *FCER1G* was replicated on 1q23.3 (z = 5.08, $P_{TWAS}$ = 3.71E−7), but not *DMPK* on 19q13.32 (z = 1.78, $P_{TWAS}$ = 0.075). Table 3 shows replication of TWAS results in GTEx lung for the 21 asthma-associated loci. For asthma loci with a single TWAS gene, consistency was observed for *RERE* on 1p36, *CLEC16A* on 16p13, and *IL4R* on 16p12. On 5q22, *TSLP* was the most significant TWAS gene in both our lung eQTL set and GTEx lung. On 17q12-q21, *GSDMB* and *ORMDL3* were switched as the most significant TWAS gene. In general, for asthma loci with multiple TWAS genes in our lung eQTL dataset (the MHC locus for example), some of the genes were replicated in GTEx lung, but the ranking of genes based on level of significance changed, and sometimes different TWAS genes were observed in GTEx lung. Four TWAS genes were replicated, but with a different direction of effect, *SMAD3* on 15q22-q23 is an example. Finally, replication was not feasible for 24 TWAS genes observed in our lung eQTL dataset as they did not yield significant gene expression prediction models in GTEx lung (Table 3).

To further filter lung TWAS genes, we used Bayesian colocalization tests for GWAS and lung eQTL signals in asthma risk loci. A high probability of shared GWAS and lung eQTL signals was observed for *GSDMB* on 17q12-q21 (PP4 = 0.84),

**Table 3 Lung TWAS genes identified in asthma-associated loci.**

| Chr band | GWAS sentinel | Lung eQTL Genes (direction, $P_{TWAS}$) | Replication in GTEx lung* Genes (direction, $P_{TWAS}$) |
|---|---|---|---|
| 1p36 | rs4480384 | **RERE** (+, 4.43E−9) | **RERE** (+, 1.69E−9) |
| 1q21 | rs558269137 | **LINGO4** (+, 6.45E−8) | **LINGO4** (+, 1.67E−6) → FLG (−, 3.16E−6) |
| 2q12 | rs72823641 | IL1RL1 (−, 4.01E−8) → **SLC9A2** (+, 1.41E−7) | **SLC9A2** (+, 1.72E−8) repl.: IL1RL1 (−, 0.163) |
| 2q37 | rs34290285 | **ING5** (−, 7.92E−8) | RTP5 (−, 4.68E−15) → D2HGDH (+, 1.15E−14) → PDCD1 (−, 9.96E−12) → **ING5** (−, 6.67E-9) → BOK (−, 2.52E−7) |
| 3q27-q28 | rs13099273 | LPP (+, 1.33E−6) | LINC01063 (−, 8.06E−15) repl.: LPP (−, 0.006) |
| 4p14 | rs5743618 | TLR1 (+, 5.15−17) | repl.: no model for TLR1 |
| 5q22 | rs1837253 | **TSLP** (+, 9.43E−14) → CAMK4 (+, 2.94E−9) → **WDR36** (−, 2.21E−6) | **TSLP** (+, 3.54E−14) → **WDR36** (−, 1.19E−10) repl.: no model for CAMK4 |
| 5q31 | rs848 | SEPT8 (−, 3.11E−23) → PDLIM4 (+, 5.54E−19) → **SLC22A5** (+, 1.03E−18) → P4HA2 (−, 6.29E−18) → SLC22A4 (+, 1.50E−17) → KIF3A (+, 9.92E−10) → HSPA4 (+, 1.98E−9) → **RAD50** (+, 5.60E−8) → CSF2 (−, 1.43E−6) | **SLC22A5** (+, 2.91E−20) → AFF4 (+, 3.16E−08) → KIF3A (−, 3.22E−07) repl.: HSPA4 (+, 0.182), **RAD50** (+, 2.63E−4), no model for SEPT8, PDLIM4, P4HA2, SLC22A4, CSF2 |
| 6p22-p21 | rs9273386 | HLA-DRB6 (+, 1.06E−20) → **HLA-DQB1** (−, 2.03E−14) → **HLA-DQB2** (−, 6.30E−14) → HLA-DPB1 (−, 2.80E−10) → **TAP2** (−, 4.01E−8) → PSMB9 (−, 2.18E−7) → TRIM10 (+, 9.57E−7) | HLA-DQA1 (−, 2.78E−51) → **HLA-DQB2** (+, 2.64E−40) → **HLA-DQB1** (−, 1.26E−39) → HLA-DQA2 (+, 1.70E−34) → HLA-DQB1-AS1 (−, 1.51E−27) → C6orf47 (−, 8.88E−17) → HLA-DRB1 (−, 1.77E−12) → ZNRD1 (−, 1.21E−8) → COL11A2 (+, 1.00E−7) → LEMD2 (+, 1.02E−6) → CFB (+, 1.37E−6) → DXO (−, 2.98E−6) repl.: HLA-DPB1 (+, 0.001), **TAP2** (−, 0.042), PSMB9 (+, 0.574), no model for HLA-DRB6, TRIM10 |
| 6q22 | rs802731 | PTPRK (+, 1.86E−10) | repl.: no model for PTPRK |
| 9p24 | rs992969 | C9orf38 (+, 1.19E−12) → KIAA1432 (+, 1.90E−6) | repl.: no model for C9orf38, KIAA1432 |
| 10p15 | rs12722502 | RBM17 (−, 2.34E−8) | repl.: no model for RBM17 |
| 10q23 | rs1870140 | TSPAN14 (−, 1.20E−6) | repl.: no model for TSPAN14 |
| 12q13 | rs3024971 | CDK2 (+, 2.54E−10) → FAM62A (+, 6.03E−8) → **RDH16** (+, 1.81E−6) | RPS26 (+, 1.29E−13) → SUOX (−, 1.21E−8) → HSD17B6 (+, 3.99E−6) repl.: **RDH16** (+, 5.36E−4), no model for CDK2, FAM62A |
| 13q32 | rs34259893 | UBAC2 (−, 1.10E−12) | repl.: no model for UBAC2 |
| 15q22-q23 | rs56375023 | SMAD3 (+, 5.41E−10) → MAP2K5 (+, 2.74E−7) | IQCH (+, 6.75E−10) → AAGAB (−, 3.37E−6) repl.: SMAD3 (−, 0.010), **MAP2K5** (+, 0.043) |
| 16p13 | rs35441874 | **CLEC16A** (+, 4.71E−9) | **CLEC16A** (+, 8.47E−07) |
| 16p12 | rs3785356 | **IL4R** (−, 5.94E−9) | **IL4R** (−, 1.00E−11) |
| 17q12-q21.2 | rs4795401 | **GSDMB** (+, 1.42E−54) → **ORMDL3** (+, 2.12E−44) → PERLD1 (+, 2.64E−26) → **GSDMA** (−, 5.52E−23) → **PNMT** (+, 7.87E−23) → **CASC3** (+, 1.53E−9) → PSMD3 (−, 4.37E−9) → **SMARCE1** (+, 6.39E−9) → CRKRS (+, 2.76E−8) → MED1 (+, 4.89E−7) → IKZF3 (−, 6.24E−7) | **ORMDL3** (+, 1.05E−54) → **GSDMB** (+, 1.82E−47) → **GSDMA** (−, 8.97E−21) → **PNMT** (+, 7.03E−20) → PGAP3 (+, 3.15E−19) repl.: **CASC3** (+, 0.005), **SMARCE1** (−, 0.003), no model for PERLD1, PSMD3, CRKRS, MED1, IKZF3 |
| 17q21.32 | rs72833417 | KPNB1 (+, 4.88E−7) | repl.: no model for KPNB1 |
| 22q13 | rs34290865 | PHF5A (+, 2.17E−8) → **MEI1** (+, 1.91E−7) | **MEI1** (+, 4.33E−8) → ACO2 (+, 1.04E−7) repl.: no model for PHF5A |

(+) and (−) indicate predicted gene expression positively or negatively associated with asthma risk. For loci with more than one TWAS genes, the genes are ordered by their level of significance and separated by arrows.
In bold are lung TWAS genes that replicated in GTEx lung.
Underlined are lung TWAS genes not reported in previous asthma GWAS.
*All Bonferroni-corrected TWAS genes per loci found in GTEx lung are indicated as well as the results of TWAS genes identified in the lung eQTL dataset in order to seek for replication ($P_{TWAS}$ < 0.05) in GTEx lung.

TLR1 on 4p14 (PP4 = 0.75), TSPAN14 on 10q23 (PP4 = 0.72), RERE on 1p36 (PP4 = 0.71), and UBAC2 on 13q32 (PP4 = 0.65) as well as two genes on 22q13: PHF5A (PP4 = 0.87) and MEI1 (PP4 = 0.63). Supplementary Data 9 shows the colocalization results for all TWAS genes identified in the lung eQTL dataset.

**Asthma TWAS in other asthma-relevant tissues in GTEx.** Asthma TWAS genes were also explored in five additional tissues in GTEx, namely blood, skin (exposed or not to sun), small intestine and spleen. Genome-wide TWAS results for all tissues are illustrated in Fig. 4. The numbers of TWAS genes reaching significance were 63 in blood (Fig. 4a), 65 for skin not sun exposed (Fig. 4b), 66 for skin sun exposed (Fig. 4c), 27 for small intestine (Fig. 4d), and 34 for spleen (Fig. 4e). Interestingly, many of these genes overlapped with those identified using lung data

(Table 3 and Fig. 4). We have also evaluated overlapping TWAS genes across GTEx tissues (Supplementary Fig. 7). Genes identified in at least four out of five tissues include those on 2q37.3 (D2HGDH), 5q31.1 (SLC22A5, KIF3A), 6p22-p21-MHC (HLA-DQA1, HLA-DQA2, HLA-DQB1, HLA-DQB1-AS1, and HLA-DRB1), 6q23.3 (AHI1), 12q13.2 (RPS26, SUOX), and 17q12-q21.2 (ORMDL3, GSDMA, GSDMB, PGAP3, MED24). Among them, SLC22A5, KIF3A, HLA-DQB1, ORMDL3, GSDMA, and GSDMB were also identified in lung. The full summary statistics for the asthma TWAS using blood, skin (sun exposed or not), small intestine and spleen are available in the Supplementary Data 22, 23, 24, 25, 26.

**Cell and tissue functional enrichment of asthma-associated SNPs.** We used GARFIELD[19] to evaluate the enrichment of

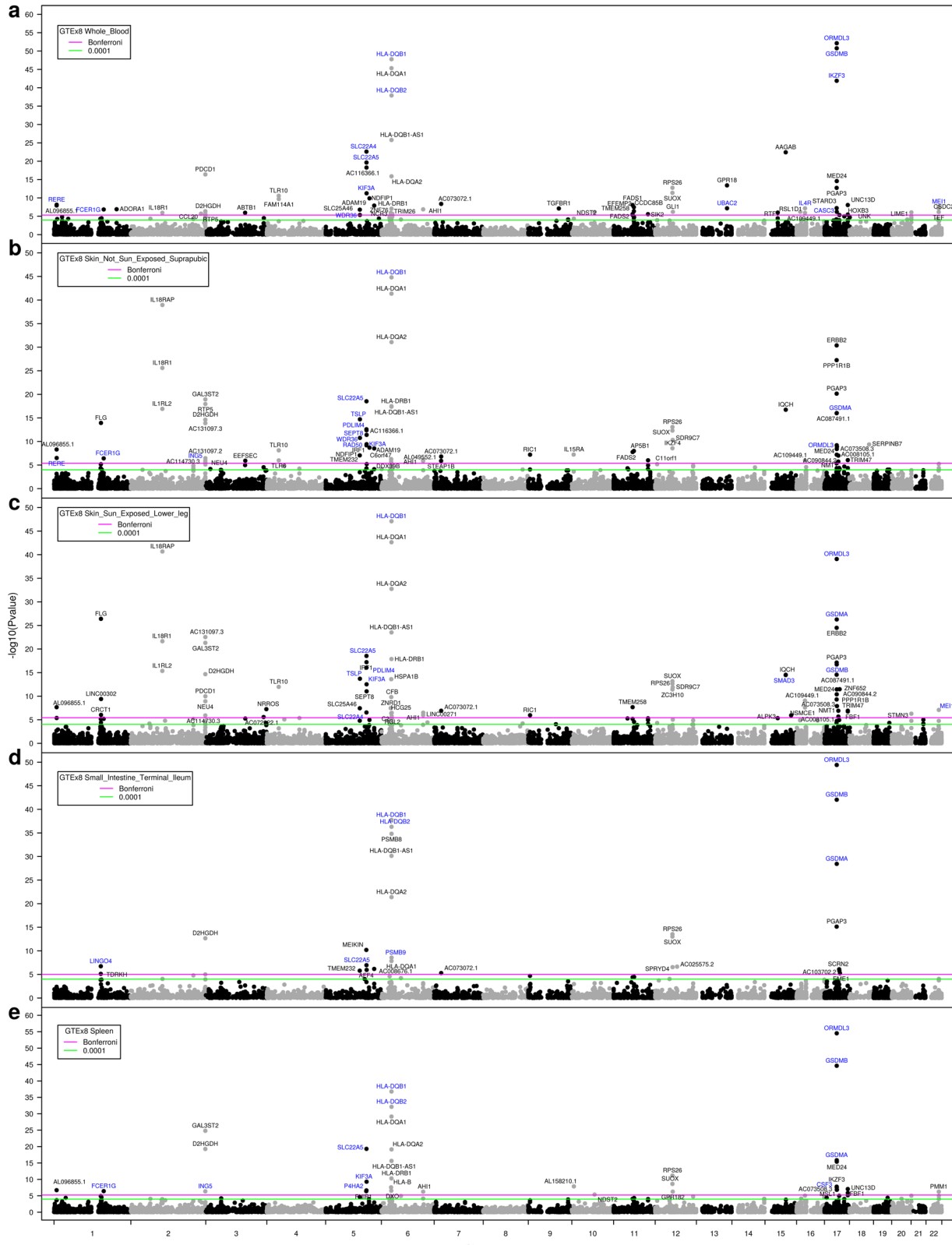

**Fig. 4 Manhattan plots of the TWAS on asthma integrating the UK Biobank GWAS and the eQTL from five tissues in GTEx.** TWAS results for **a** in blood, **b** in skin not sun exposed, **c** in skin sun exposed, **d** in small intestine and **e** in spleen are illustrated. Each dot represents the association between predicted gene expression and asthma for a specific gene/transcript. $P$ values for gene expression-asthma associations are on the $y$ axis in $-\log 10$ scale. The green and magenta horizontal lines represent $P_{TWAS}$ of 0.0001 and Bonferroni, respectively. Annotations for genome-wide significant genes/transcripts that passed Bonferroni correction are indicated. Genes in blue have also been identified in lung tissue.

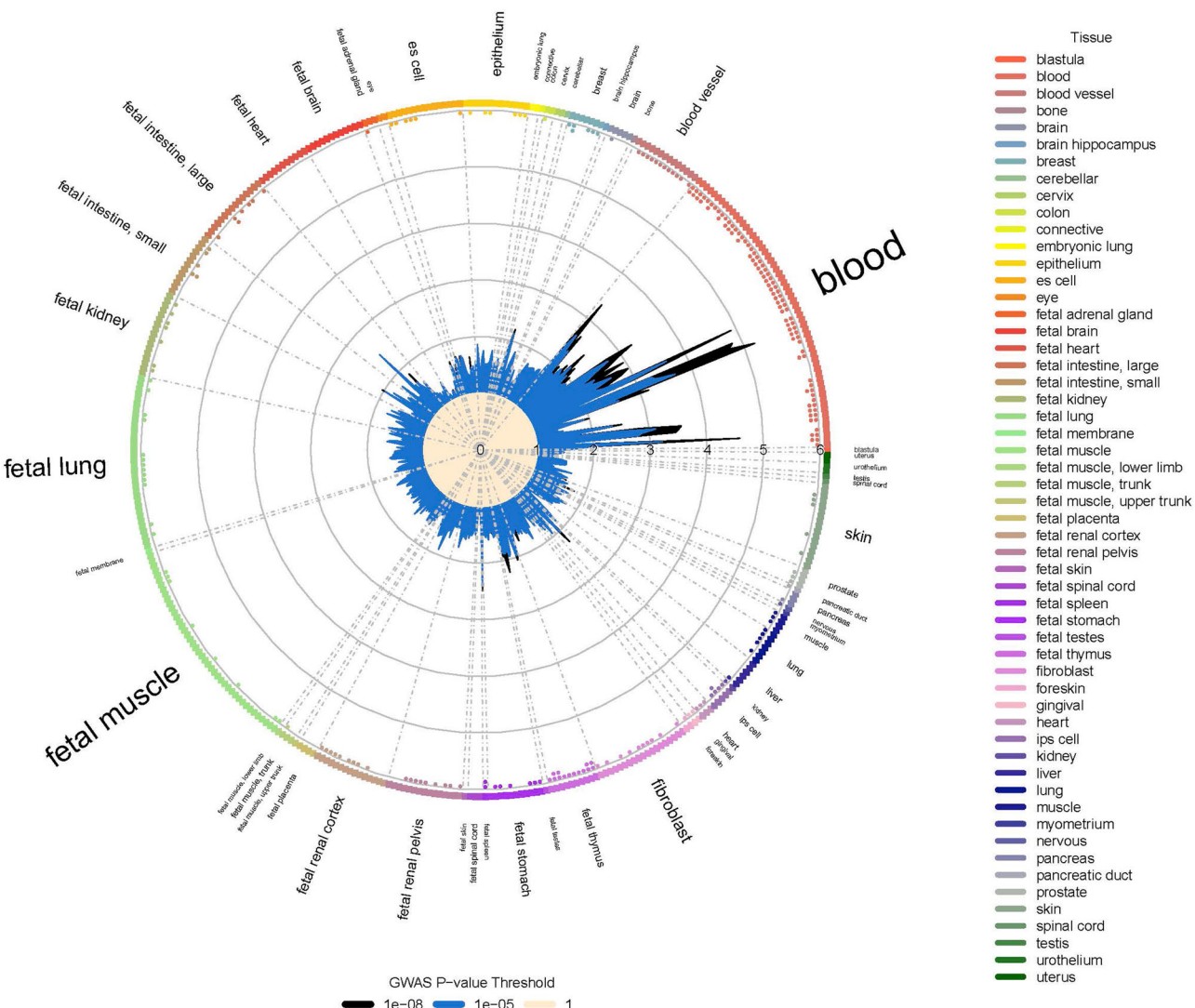

**Fig. 5 GARFIELD functional enrichment analyses.** The wheel plot shows functional enrichment for asthma variants within DNase I hypersensitivity site hotspot regions in ENCODE and Roadmap Epigenomics data. The radial axis represents the enrichment (OR) for each of 424 cell types that are sorted by tissue along the outside edge of the plot. Tissues are labeled with font size proportional to the number of cell types. Boxes forming the edge are colored by tissue. Enrichment is calculated for two GWAS significance thresholds: 1.0E−5 and 1.0E−8, which are plotted in blue and black, respectively, inside the plot. Dots along the inside edge of the plot are colored with respect to tissue and represent significant enrichment for a specific cell type (one dot = $P < 1E−5$ and two dots = $P < 1E−6$).

asthma-associated loci in regulatory and functional annotations derived from ENCODE and the Roadmap Epigenomics Project. Figure 5 shows functional enrichment within DNase I hypersensitivity site hotspots at two GWAS P value cut-offs. The largest fold enrichment values were in the blood. All results are summarized in Supplementary Data 10, along with other annotation types.

**Functional mapping and annotation in blood.** Considering the strong enrichment of genetic association data for asthma in blood cells including B and T cells, we leveraged blood *cis*-eQTL from 31,684 samples[9] to identify genetically expressed genes (eGenes) associated with asthma. In total, we identified 128,752 significant SNP-gene pairs ($P_{FDR} < 0.05$), which mapped 485 blood eGenes (Supplementary Data 11 and 12). The blood eGenes associated with asthma were enriched in KEGG for antigen processing and presentation ($P_{adjusted} = 1.74E−23$) and Th17 cell differentiation ($P_{adjusted} = 6.60E−22$) (Supplementary Data 13). As GARFIELD showed a strong enrichment of gene variants associated with

asthma in blood and GM12878 cell line (lymphoblastoid B cell line) (Fig. 5 and Supplementary Data 10), we mapped individual significant SNPs ($P_{GWAS} < 5.0E−8$, $r^2 < 0.6$) (see "Methods" section) to genes by using Hi-C data obtained in this cell line. Chromatin contact mapping in GM12878 identified 563 genes (Supplementary Data 14), which included 251 blood eGenes. Although interesting, prioritizing candidate genes for asthma was still challenging at many loci. For example, Fig. 6 shows a zoom in of a circos plot for genetic association data integrating eQTL and chromatin contact mappings in GM12878 at the 17q12-q21 locus. Many eQTL genes (in green), Hi-C genes (in orange) or genes significant in both eQTL and Hi-C data (in red) were identified. We have thus decided to perform additional filtering using Mendelian Randomization.

**Mendelian randomization in blood with asthma.** We next implemented two-sample Mendelian Randomization to infer causal associations between blood eGenes and asthma (see "Methods" section). We were able to perform 431 Mendelian

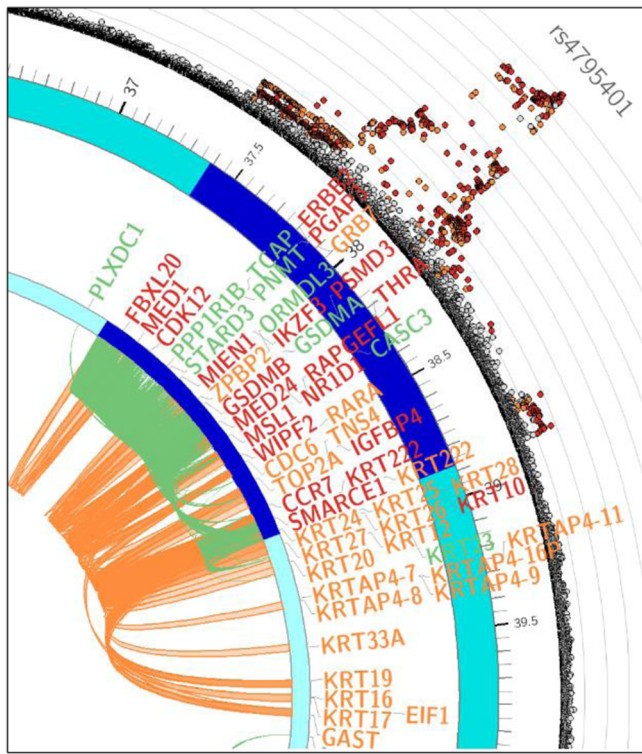

**Fig. 6 Blood eQTLs and chromatin interactions in GM12878 at the 17q12-q21 asthma risk locus.** The most significant asthma GWAS SNP is indicated at the most outer border of the circos plot. The subsequent layers show 1) genetic association results from the asthma GWAS in UK Biobank for SNPs with P value < 0.05 with the color of dots reflecting the level of LD with the sentinel variant (red: $r^2 > 0.8$, orange: $r^2 > 0.6$), 2) the chromosome coordinate and the asthma GWAS loci highlighted in dark blue, 3) blood eQTL genes, genes mapped by Hi-C, and genes mapped by both eQTL and Hi-C labeled in green, orange, and red, respectively, 4) green and orange lines link the position of eQTLs and chromatin interactions, respectively.

Randomizations with at least three instrumental variables per gene (mean of 19 instrumental variables per gene) (Supplementary Data 15) and we identified 50 blood eGenes at a Bonferroni threshold ($P_{IVW} = 0.05/431 = 1.16E-4$) that were causally associated with asthma (Supplementary Data 16). Supplementary Fig. 8 shows the most significant causally associated blood eGene identified per asthma-associated loci. Among the 50 blood eGenes, 26 did not show heterogeneity on the Cochrane's Q-test ($P > 0.05$), whereas 24 eGenes had heterogeneity ($P_{Q-test} < 0.05$). These 24 eGenes were corrected by applying the MR-PRESSO approach (methods) and all remain significant (Supplementary Data 16). At 17q12-q21, the expression of *GSDMB* (per SD OR: 1.11, 95% CI: 1.09–1.14, $P_{MR\_Corr} = 2.14E-13$) was positively associated with the risk of asthma (Supplementary Data 16). Among the causally associated eGenes, 17 showed colocalization signals (PP4 > 0.6) between blood eQTLs and genetic association data (*RERE, EFEMP2, TGFBR1, UBAC2, SIK2, UNC13D, TLR10, LGSN, MEI1, AHI1, CXCR5, FADS1, FADS2, TEF, CSDC2, GSDMB*, and *IKZF3*) (Supplementary Data 16).

**Drug targets.** Target genes of the asthma-associated variants identified in previous sections were then integrated to prioritize druggable genes. In total, we identified 55 lung TWAS genes (Supplementary Data 7), 485 blood eGenes (Supplementary Data 12), and 563 chromatin contacts mapped genes (Supplementary Data 14). Together, 806 unique target genes were

identified with overlap across methods shown in Supplementary Fig. 9. Notably, 101 of them overlapped with the recent list of 161 possible asthma drug targets summarized by El-Husseini et al.[20], which were derived from eQTL and non-synonymous analysis of independent variants associated with asthma in previous GWAS (Supplementary Data 17). According to the Open Targets Platform[21], 13 out of 806 are the targets of investigational or approved asthma drugs (Supplementary Table 1). All target genes were also interrogated using the Open Targets Platform[21] for their overall association score with asthma. Results for all target genes are in Supplementary Data 17. The 806 target genes were also overlaid with the known druggable genes derived from the drug–gene interaction database (DGIdb)[22] and the druggable genome[23]. Drug–gene interactions were identified for 182 target genes in DGIdb and 201 target genes were part of the druggable genome (Supplementary Data 17), which offer numerous opportunities for drug repurposing. We further focused on 29 target genes that were consistently identified by TWAS, eQTL, and chromatin interactions (Supplementary Fig. 9). Ten of them have known drug targets. Supplementary Table 2 shows these ten druggable target genes for asthma and their direction of effect on asthma risk in lung tissue as well as the candidate drugs, interaction types and clinical indications. Target-asthma associations of 1, which is the highest possible score in Open Targets, were observed for two genes including *IL4R* that is the therapeutic target of dupilumab used to treat uncontrolled persistent asthma[24] and *SMAD3* involved in airway remodeling[25] and that may mediate some actions of corticosteroids, which are the cornerstone of asthma treatment. Finally, we filtered the 806 target genes based on two cumulative criteria: 1) those with asthma score of at least 0.5 in Open Targets, and 2) those that are druggable in either the DGIdb or the druggable genome. By excluding the HLA molecules, this strategy revealed 40 prioritized therapeutic targets for asthma (Table 4). In addition to *IL4R* and *SMAD3*, these prioritized genes are known targets of existing asthma drugs including *IL6* (clazakizumab, sirukumab), *TNFSF4* (oxelumab), *TSLP* (tezepelumab), *CCR4* (mogamulizumab), *IL13* (anrukinzumab, lebrikizumab, dectrekumab, and tralokinumab), *IL5* (mepolizumab, reslizumab), and *IL2RA* (daclizumab).

Finally, we performed a cross-phenotype search in GeneATLAS[26] to evaluate potential effects of modulating genes prioritized as therapeutic targets for asthma. Diseases/traits from UK Biobank participants that are significantly associated with the GWAS sentinel variants underpinning the 40 prioritized target genes are indicated in Supplementary Data 18. Based on this in silico approach and available data, the safety profile of most of these targets seems favorable with genetic associations observed mostly with asthma-related phenotypes, white blood cells (eosinophil, neutrophil, and lymphocyte), and other allergic conditions (hay fever/allergic rhinitis and allergy/hypersensitivity/anaphylaxis). Targets located in the MHC locus (*NOTCH4* and *ITPR3*) were associated with 41 different phenotypes, including ulcerative colitis and multiple sclerosis signals in the opposite direction from asthma. Similarly, the lead variant at the *IL5/IL13* locus linked with a decreased risk of asthma was associated with an increased risk of psoriasis.

## Discussion

An important genetic susceptibility to develop asthma has long been demonstrated by genetic epidemiology studies[27]. However, the predisposing genetic variants have been difficult to identify until the realization of recent large-scale GWAS. Now, a large number of genetic loci are robustly associated with asthma. The new challenge is to identify the candidate causal genes and best therapeutic targets underpinning GWAS-nominated loci. Here, we leveraged lung and blood transcriptome as well as epigenetic

**Table 4 Genes prioritized as therapeutic targets for asthma in alphabetic order.**

| Genes | Lung TWAS gene | Blood eGene | Hi-C gene | Asthma score[a] | DGIdb 3.0[b] | Druggable genome[c] | Asthma drug targets[d] |
|---|---|---|---|---|---|---|---|
| *CCR4* | No | Yes | No | 0.62 | Yes | Yes | No |
| CCR7 | No | Yes | Yes | 0.55 | No | Yes | Yes |
| CD247 | No | Yes | Yes | 1 | Yes | Yes | Yes |
| ERBB2 | No | Yes | Yes | 0.51 | Yes | Yes | No |
| ERBB3 | No | Yes | Yes | 0.62 | Yes | Yes | Yes |
| FADS2 | No | Yes | No | 0.63 | Yes | No | Yes |
| FASLG | No | No | Yes | 0.84 | No | Yes | No |
| FCER1G | Yes | No | No | 0.64 | Yes | No | Yes |
| GLB1 | No | Yes | Yes | 0.71 | No | Yes | No |
| GPR18 | No | Yes | Yes | 0.50 | Yes | Yes | No |
| GPR183 | No | Yes | Yes | 0.54 | Yes | Yes | Yes |
| *IL13* | No | No | Yes | 1 | Yes | Yes | No |
| IL18R1 | No | Yes | No | 1 | No | Yes | No |
| IL18RAP | No | Yes | No | 0.57 | No | Yes | Yes |
| IL1RL1 | Yes | Yes | No | 1 | No | Yes | Yes |
| *IL2RA* | No | Yes | No | 1 | Yes | Yes | Yes |
| IL33 | No | No | Yes | 1 | No | Yes | No |
| *IL4R* | Yes | Yes | Yes | 1 | Yes | Yes | Yes |
| *IL5* | No | No | Yes | 1 | Yes | Yes | No |
| *IL6* | No | Yes | Yes | 1 | Yes | Yes | No |
| IL7R | No | Yes | Yes | 1 | Yes | Yes | Yes |
| ITGB8 | No | No | Yes | 1 | No | Yes | Yes |
| ITPR3 | No | Yes | No | 0.73 | Yes | Yes | Yes |
| MYC | No | Yes | No | 1 | Yes | No | No |
| NDFIP1 | No | Yes | No | 0.80 | No | Yes | Yes |
| NOTCH4 | No | Yes | Yes | 0.65 | Yes | Yes | No |
| PLXNC1 | No | Yes | Yes | 1 | No | Yes | No |
| PRKCQ | No | Yes | No | 1 | Yes | Yes | Yes |
| PTPRC | No | Yes | Yes | 1 | Yes | Yes | No |
| RORA | No | Yes | No | 1 | Yes | Yes | No |
| RORC | No | Yes | Yes | 0.67 | Yes | Yes | No |
| RUNX1 | No | Yes | No | 0.80 | Yes | No | No |
| *SMAD3* | Yes | Yes | Yes | 1 | Yes | Yes | No |
| STAT6 | No | Yes | No | 1 | Yes | Yes | Yes |
| TLR1 | Yes | Yes | Yes | 1 | No | Yes | Yes |
| TLR10 | No | Yes | Yes | 0.94 | No | Yes | No |
| TLR6 | No | Yes | Yes | 0.77 | No | Yes | Yes |
| TNFRSF8 | No | No | Yes | 1 | Yes | Yes | No |
| *TNFSF4* | No | Yes | Yes | 1 | Yes | Yes | Yes |
| *TSLP* | Yes | No | Yes | 1 | No | Yes | Yes |

Targets of existing asthma drugs are in bold.
[a]Overall association score for asthma from the Open Targets Platform[21].
[b]DGIdb, Drug-gene interaction database[22].
[c]Druggable genome[23].
[d]Asthma drug targets derived from El-Husseini et al.[20]

marks to identify the most likely causal genes within asthma susceptibility loci derived from UK Biobank. Using a broad asthma definition, we identified 72 physically-defined asthma loci containing 116 independent genetic variants with $P_{GWAS} < 5.0E -8$. The effect size estimates were robust to more strict asthma definitions excluding other lung diseases, smoking history or allergy within controls. As expected, the yield of deleterious coding variants was low (eight nonsynonymous variants and one stop-gain variant with CADD score >20), and we thus focused most analyses on regulatory elements. The UK Biobank asthma GWAS was integrated with the largest lung eQTL study available. Fifty-five significant TWAS genes located in 21 previously reported asthma loci and two novel asthma risk loci (1q23.3-*FCER1G* and 19q13.32-*DMPK*) were found and 23 of them (in 14 loci) replicated in GTEx lung. As previously reported[28], we demonstrated a strong enrichment of asthma-associated variants in regions of regulatory and functional annotations in blood. We mapped 485 blood eGenes and demonstrated that 50 of them are causally associated with asthma by Mendelian randomization.

Chromatin contact mapping in the blood cell line showing the most significant enrichment of DNase I hypersensitive sites (GM12878) revealed 563 Hi-C genes. Prioritization of the 806 candidate causal genes identified in this study based on consistency across methods, druggability, and prior asthma association led to 40 genes prioritized as therapeutic targets for asthma. We performed a quick PheWAS lookup on prioritized genes to provide a first appreciation of any potential side effects of targeting these genes. Although a more in-depth investigation will be needed on each gene, the direction of effect on asthma compared to other health conditions and most particularly allergic phenotypes seem to be concordant. Exceptions included genes located at the MHC locus showing a potential increase in risk of ulcerative colitis and multiple sclerosis, and at the *IL5/IL13* locus showing a potential increase in risk of psoriasis.

Nine of the 40 prioritized genes are the targets of existing asthma drugs including *IL4R* (dupilumab), *SMAD3* (corticosteroids), *IL6* (clazakizumab, sirukumab), *TNFSF4* (oxelumab), *TSLP* (tezepelumab), *CCR4* (mogamulizumab), *IL13* (anrukinzumab,

lebrikizumab, dectrekumab, tralokinumab), *IL5* (mepolizumab, reslizumab), and *IL2RA* (daclizumab). This supports the possibility that other genes in that list (Table 4) are credible therapeutic targets for asthma. Among them, there are three members of the toll-like receptor family: *TLR1*, *TLR6*, and *TLR10*. These three TLRs are located at the same 4p14 locus, are phylogenetically related, and require the formation of heterodimers with TLR2 for recognition of invading microbes[29]. We found a strong colocalization signal between the blood eQTL for *TLR10* and the GWAS for asthma (PP4 = 0.84). In Mendelian randomization, the blood expression of *TLR10* was positively associated with the risk of asthma ($P_{IVW}$ = 4.34E−6). We also identified two missense mutations in *TLR10*, C-rs4129009 (p.Ile775Val) ($P_{GWAS}$ = 1.49E−10) and G-rs11096957 (p.Asn241His) ($P_{GWAS}$ = 2.49E−10), which are in moderate LD ($r^2$ = 0.42 in CEU) and associated with the risk of asthma. The CADD score for G-rs11096957 is 21.9 and is predicted to be "possibly damaging" and "deleterious" by PolyPhen and SIFT, respectively. The alleles C-rs4129009 (p.Ile775Val) and G-rs11096957 (p.Asn241His), which decrease the risk of asthma, have been associated with elevated blood cytokine responses to a TLR1/2 agonist, most specifically $Pam_3CSK_4$-induced interleukin 6 (IL6)[30]. We recently demonstrated that genetically predicted levels of circulating IL6R, a negative regulator of IL6 signaling, are positively associated with the risk of asthma and atopic disorders[31]. These data suggest a complex interaction between TLR10 and IL6 on the risk of asthma and warrant further investigation. One line of inquiry could examine the possibility that TLR10 dampens TLR2 signaling and IL6 production, thereby increasing the risk of asthma. In support of the latter hypothesis, in an ovalbumin-induced asthma mouse model, IL6 lowered Th2 cytokines and decreased bronchial hyperresponsiveness[32].

Other gene targets include cytokine/chemokine receptors (*IL7R*, *IL1RL1*, *IL33*, *IL18R1*, *IL18RAP CCR7*), members of the EGFR family of receptor tyrosine kinases (*ERBB2* and *ERBB3*), protein kinase C theta (*PRKCQ*), G protein-coupled receptors (*GPR18* and *GPR183*), an antigen recognition molecule (*CD247*), a member of the protein tyrosine phosphatase family (*PTPRC*), transcription factors (*STAT6*, *RORC*, *RUNX1*, *RORA*, and *MYC*), a member of the NOTCH family (*NOTCH4*), TNF receptor superfamily member 8 (*TNFRSF8*), Fas ligand (*FASLG*), a member of the plexin family (*PLXNC1*), integrin subunit beta 8 (*ITGB8*), Nedd4 family interacting protein 1 (*NDFIP1*), inositol 1,4,5-trisphosphate receptor type 3 (*ITPR3*), fatty acid desaturase 2 (*FADS2*), and galactosidase beta 1 (*GLB1*). All these targets have drug–gene interactions in DGIdb[22] and/or are present in the list of genes encoding druggable human proteins[23]. They are thus representing drug repurposing/development opportunities. Further experimental research will be needed to screen these putative novel therapeutic targets for asthma.

The effect sizes of all genetic variants associated with asthma are relatively small. Although 116 independent variants reached genome-wide significance ($P_{GWAS}$ < 5.0E−8) in this study, the ORs range from 1.03 to 3.59 (median = 1.06, note that ORs lower than 1 were converted into their reciprocal (1/OR)). As an example to appreciate the effect size that we are detecting, the most significant associated asthma variant on chromosome 9 near the *IL33* gene has a *P* value of 1.25E−56 and an OR of 1.13, which is the result of allele frequencies of 0.73 in cases and 0.75 in controls. One hundred out of the 116 independent variants were common, with minor allele frequency greater than 5% in cases and controls combined. So most risk alleles are common with small effects that we are able to detect owing to the large sample size. Cumulatively, all genetic variants discovered by GWAS explained about 8–9% of the total heritability[28], suggesting much more work is needed to elucidate the full genetic architecture of

asthma. More work is also needed to move discovered genetic factors underlying asthma down the clinical translation pipeline. We believe that the current study represents an important step beyond GWAS data. By combining different data sources (eQTL and Hi-C in disease-relevant tissues) and advanced bioinformatics approaches (TWAS, Mendelian Randomization, colocalization), we were able to reveal relevant genes and putative therapeutic targets for asthma.

As observed in previous asthma GWA studies, we had limited success in mapping asthma-associated variants to deleterious coding SNPs. One of our most interesting hit is with the filaggrin (*FLG*) gene. *FLG* is located on chromosome 1q21.3, a locus where we have identified five independent significant variants, which have all been previously reported (Supplementary Data 2 and 3). Two of the five independent variants are changing the structure of the protein including the sentinel variant rs558269137 (p.Ser761CysfsX36) and variant rs61816761 (p.Arg501Ter). *FLG* was previously associated with atopic dermatitis[33] where the risk variants are believed to disrupt the skin barrier, allowing allergen sensitization and then promoting the development of asthma[34]. rs61816761 was also found in previous GWAS of asthma in UK Biobank[6,7,28,35,36] and with greater effect on atopic dermatitis than asthma[37], which supports skin barrier dysfunction as a cause of asthma. Another deleterious coding variant (rs2230624, Cys273Tyr) was identified in *TNFRSF8* (also known as *CD30*), which was previously reported[38] and characterized as a loss of function variant that decreased asthma risk by reducing the trafficking of the CD30 protein on cell surface[37].

As GWAS on asthma in UK Biobank accumulate[6,7,28,36–39], the next milestone will be to identify the function units, most intuitively genes, underlying the GWAS loci. In this study, we have combined the UK Biobank GWAS data with the largest lung eQTL available to perform a TWAS. Plausible causal genes in lung tissues were revealed for 21 asthma loci. On 17q12-q21.2, the first discovered[40] and the most replicated[41] GWAS asthma locus, *GSDMB* was the most significant TWAS gene in our lung eQTL dataset. Although other TWAS genes were observed in that locus, the *P* value distributions of GWAS and eQTL colocalized better with *GSDMB*. Using blood as eQTL source, we also identified *GSDMB* as the most likely causal gene on 17q12-q21.2. These results are consistent with eQTL analysis showing that SNPs associated with asthma susceptibility and severity at 17q12-q21.2 are correlated with *GSDMB* expression in cells from human bronchial epithelial biopsy and bronchial alveolar lavage[42,43]. *GSDMB* is thus a gene to focus on in future functional studies. Other lung TWAS genes prioritized by our study for further functional characterization are *RERE* on 1p36, *TLR1* on 4p14, *SLC22A5* or *RAD50* on 5q31, *RBM17* on 10p15, *UBAC2* on 13q32, *SMAD3* on 15q22-q23, *CLEC16A* on 16p13, *IL4R* on 16p12, *KPNB1* on 17q21.32, and *PHF5A* on 22q13.

The lung TWAS revealed a novel asthma risk locus at 1q23.3 with the putative gene encoding the gamma chain of the high-affinity IgE receptor (*FCER1G*). Note that the *FCER1A* gene at 1q23.2 (approximately 2 Mb away from *FCER1G*) has been associated with total serum IgE levels[44]. Concerning *FCER1G*, hypomethylation at its promoter has been reported in monocytes of patients with atopic dermatitis, resulting in the overexpression of high affinity IgE receptors in these cells[45]. Here, we found that higher expression of *FCER1G* in lung tissue is associated with asthma ($z$ = 4.74, $P_{TWAS}$ = 2.13E−6), a finding replicated in GTEx lung ($z$ = 5.08, $P_{TWAS}$ = 3.71E−7). This new asthma locus may thus mediate its effect by upregulating *FCER1G*, which may then lead to inflammatory cells with greater surface expression of IgE receptors, that are more prone to allergic reaction. *FCER1G* was also a significant TWAS gene in other asthma-relevant tissues including blood, skin and spleen.

Replication in this study is challenging as we have used the largest asthma GWAS study and the largest lung eQTL study. Similarly powered replication sets are currently not available. For the lung TWAS genes, our best attempt to replicate the novel findings was to use the lung eQTL set from GTEx. Unfortunately, 24 out of 55 genes identified using the lung eQTL study did not yield significant gene expression prediction models and thus could not be evaluated by TWAS. This is partly the results of a smaller sample size in GTEx lung ($n = 515$ vs. 1038), but many other factors. It should be emphasized that a head-to-head comparison between two eQTL sets is not straightforward as two GWAS sets and this not just because of the nature of data, i.e., static for SNP compared to dynamic and cell/tissue dependent for gene expression. There are major differences between our lung eQTL study and GTEx lung including the human lung sampling (surgery vs. post-mortem) and the gene expression platforms (microarray vs. RNA-Seq). Tissue processing methods for freezing, storing, and thawing tissues as well as extracting RNA were also different. Other investigators have also highlighted extensive heterogeneity in gene expression for the lung transcriptome data in GTEx due to sampling location in the lung and treatment related changes such a mechanical ventilation[46]. Taken all together, GTEx lung is not the most suitable replication set, but our best option now. Despite all this, we were pleasantly surprised to replicate 23 TWAS genes in GTEx lung.

This study has limitations. We used the best possible bioinformatics approaches to identify causality genes. However, our study does not provide a complete package to support novel therapeutic hypotheses[47]. The candidate gene targets identified in our study will need to be experimentally validated by other preclinical models (cellular, animal, and human studies) in order to understand the biological effects of risk alleles on gene function and the role of these genes in the pathogenesis of asthma. Further studies are needed to demonstrate causality. For the asthma GWAS, we have tested 35,270,583 SNPs (minor allele frequency >0.0001) and used the conventional common-variant significance $P$ value threshold of $5 \times 10^{-8}$. By using a more stringent threshold of $5 \times 10^{-9}$ recommended for whole-genome sequencing studies including rare variants[48], 18 out of 72 loci reported in our study would no longer be significant. Among them, 12 of these loci have been reported in previous asthma GWAS. All 18 loci have minor allele frequency >0.05 including the six novel loci with minor allele frequency ranging from 0.15 to 0.44. We did not observe replication of these novel loci in the Trans-National Asthma Genetic Consortium[4], and they should thus be interpreted with caution. Our analyses are largely based on European-descent individuals. Inference to other ethnic groups is thus a concern and the lack of similarly powerful resources (e.g., UK Biobank) for other ancestries represent a missed opportunity to identify other relevant asthma genes. Environmental risk factors and the specific period of exposures during the lifespan play an important role in the development of asthma. Our genomic datasets (GWAS and eQTL) are retrospective in nature and mostly derived from adult populations. Environmental and age-related modifiers of genetic risk and gene expression levels are likely to have been missed. We used whole lung and blood tissues, which contain heterogeneous cell populations, limiting our ability to identify genes affecting asthma risk through gene regulation and epigenetic marks. Progress in single-cell transcriptomic is promising for future studies. Finally, we used regulatory and functional annotations derived from ENCODE and Roadmap Epigenomics data to find cell type and tissue enrichment of asthma-associated loci. Although valuable, these publicly available datasets lack functional data on some cell types that are relevant to asthma such as eosinophils and airway smooth muscle cells, which limited our ability to understand the functional impact of asthma-associated variants.

In conclusion, this study expands our understanding of the regulatory and functional mechanisms underlying GWAS asthma risk loci in lung and blood tissues. The candidate causal genes identified are key to understand disease etiology, interpret GWAS results, and prioritize follow-up functional studies. Our top therapeutic targets represent new opportunities for drug repositioning and testing in pre-clinical models.

## Methods

**Genome-wide association study on asthma in UK Biobank.** UK Biobank is an open access resource of nearly 500,000 participants enrolled at the age of 40–69 and prospectively evaluated for a range of health-related outcomes[49]. The definition of asthma in this study is based on the UK Biobank Outcome Adjudication Group and relies on hospital, death, primary care, and self-reported related codes (Phase 2 code list for asthma) (Supplementary Data 19). Asthma cases include patients with a diagnosis from hospital record (ICD-9 or ICD-10 codes) or primary care medical record as well as those with self-reported asthma (data-field 20002 in UK Biobank). Genotyping data are derived from the Affymetrix UK BiLEVE or UK Biobank Axiom Arrays. Phasing and imputation were performed centrally using the Haplotype Reference Consortium and merged UK10K and 1000 Genomes phase 3 reference panels[50,51]. Samples with call rate <95%, outlier heterozygosity rate, sex mismatch, non-white British ancestry, samples with excess third-degree relatives (>10), or not used for relatedness calculation were excluded. Variants with an imputation quality score (INFO) ≤ 0.3 or minor allele frequency <0.0001 were removed. Using the aforementioned definition of asthma and quality control filters, 56,167 asthma cases were compared to 352,255 controls. The genetic association analysis was performed using SAIGE (Scalable and Accurate Implementation of GEneralized mixed model, version 0.36.3.1, https://github.com/weizhouUMICH/SAIGE)[52]. SAIGE is a two-step method to perform generalized mixed model GWAS analysis that is robust to unbalanced case-control ratios, sample relatedness and low-frequency variants. In step 1, we fit a null logistic mixed model with 93,511 independent, high-quality genotyped variants, which were used by the UK Biobank data group to estimate the kinship coefficients between samples[51,52]. The following covariates were added: age, sex, and the first 20 ancestry-based principal components. In step 2, we performed association tests between each genetic variant (genotyped and imputed) and asthma. We applied the leave-one-chromosome-out (LOCO) scheme (LOCO = TRUE). The quantile–quantile plot was generated (Supplementary Fig. 2). The genomic inflation factor was computed by converting $P$ values into chi-squared values, and then dividing the median of the resulting chi-squared statistics by the expected median of the chi-squared distribution. The present analyses were conducted under UK Biobank data application number 25205. The study was approved by the Institut universitaire de cardiologie et de pneumologie de Québec—Université Laval (IUCPQ-UL) ethics committee.

**Heritability.** LD-score regression was used to estimate SNP-heritability for asthma[53]. To obtain heritability on the liability scale, we provided sample and population prevalence of 13.8% (—samp-prev 0.138) and 15.6%[16] (—pop-prev 0.156), respectively.

**Number of loci associated with asthma.** After the GWAS analysis, we assessed the number of loci that were associated with asthma based on two methods. First, we counted the number of loci based on physical distance only. All SNPs associated with asthma ($P < 5.0E-8$) were ranked by chromosome order and by position on build 37. Two subsequent SNPs on this list located on the same chromosome and separated by more than 1 Mb were considered distinct loci. The physical boundaries of asthma-associated loci were then defined by adding 500 Kb downstream and upstream of the most 5′ and 3′ asthma-associated variants ($P_{GWAS} < 5.0E-8$), respectively, within each locus. One exception was the extended MHC region on chromosome 6 that was counted as a single locus and delimited at 25,726,000–33,378,000 bp (GRCh37) based on the positions of two genes (HIST1H2AA and KIFC1). Second, we identified the number of independent variants, as some physically defined loci will contain significant SNPs that are not in LD. This was performed using a stepwise conditional analysis (GCTA—cojo-slct)[54] using UK Biobank as the LD reference panel. The procedure consists of a first round of analysis that is conditioned on the most significant asthma-associated variant at each locus derived from the original GWAS. If significantly associated variants remain, a second round of analysis is conditioned on the most significant asthma-associated variant from the first round. Subsequent rounds are carried out until no more variants reach $P_{GWAS} < 5.0E-8$.

**GWAS sensitivity analysis.** GWAS-nominated loci were re-evaluated by changing exclusion criteria to define asthma cases and controls. The rationale was to evaluate the potential confounding effect of other lung diseases, smoking, and allergy. Genetic association analyses were thus performed in three case–control subsets. First, asthma cases and controls with other lung diseases were excluded. Individuals were excluded if they had self-reported or medical records consistent with the presence of chronic obstructive pulmonary disease, emphysema, chronic

bronchitis, interstitial lung disease, or alpha-1 antitrypsin deficiency. This results in the exclusion of 20,998 individuals and genetic analysis performed in 47,391 asthma cases and 340,033 controls. Second, we excluded all asthma cases and controls with a positive smoking history (i.e., former and current smokers). This results in the exclusion of 250,739 individuals and genetic analysis performed in 21,097 asthma cases and 136,586 controls. Third, we excluded control individuals with atopy, including hay fever, allergic rhinitis, and eczema/atopic dermatitis. This results in the exclusion of 84,113 individuals and genetic analysis with 268,142 controls (and the same number of asthma cases as the main analysis, $n = 56,167$). Note that the three lists of exclusion criteria were applied separately (not cumulatively) and specific UK Biobank data fields and codes used for excluding individuals in each case–control subset are provided in Supplementary Data 19.

**The lung expression quantitative trait loci**. The lung eQTL dataset consists of whole-genome genotyping (Illumina Human1M-Duo BeadChip) and gene expression (Affymetrix) in non-tumor lung tissues from patients who underwent lung surgery at three academic sites, Laval University, University of British Columbia, and University of Groningen, henceforth referred to as Laval, UBC, and Groningen, respectively. All lung specimens from Laval were obtained from patients undergoing lung cancer surgery and were harvested from a site distant from the tumor. At UBC, the majority of samples were from patients undergoing resection of small peripheral lung lesions. Additional samples were from autopsy and at the time of lung transplantation. At Groningen, the lung specimens were obtained at surgery from patients with various lung diseases, including patients undergoing therapeutic resection for lung tumors, harvested from a site distant from the tumor, and lung transplantation. Lung tissue processing and storage, DNA and RNA extraction, genotyping, microarray-based gene expression, and lung *cis*-eQTL analyses have been described previously[8,55]. Following standard microarray and genotyping quality controls, data on 1038 patients are available. The demographic and clinical characteristics of the subjects are described in Supplementary Data 20. At Laval and UBC, written informed consent was obtained from all subjects and the study was approved by their respective ethics committee. At Groningen, lung specimens were provided by the local tissue bank of the Department of Pathology and the study protocol was consistent with the Research Code of the University Medical Center Groningen and Dutch national ethical and professional guidelines ("Code of conduct; Dutch federation of biomedical scientific societies"; http://www.federa.org).

**Transcriptome-wide association study (TWAS)**. The TWAS was performed using S-PrediXcan[15]. The lung eQTL dataset was used as the training set to derive the expression weights. Gene expression normalized for age, sex, and smoking status from Laval, UBC, and Groningen were combined with ComBat[56]. Gene expression traits were then trained with elastic net linear models (alpha = 0.5, n_k_folds = 10, window = 500 Kb). Models with false-discovery rate (FDR) < 0.05 as implemented in S-PrediXcan were obtained for 19,918 probe sets. Predicted expression levels from the lung in the UK Biobank participants were then tested for association with asthma[15]. The Bonferroni correction was used to claim transcriptome-wide significance (S-PrediXcan $P_{TWAS} = 0.05/19,918 = 2.51E−6$).

**Pathway analysis**. Pathway analysis was performed using the Enrichr web server[57]. Lung TWAS genes and blood eGenes discovered in this study were uploaded in Enrichr and enrichment was assessed using the combined score method for gene sets available in the Kyoto Encyclopedia of Genes and Genomes (KEGG) database.

**TWAS replication in GTEx lung**. Lung eQTL data from 515 individuals available in the Genotype-Tissue Expression (GTEx) project (version 8)[10] were used for TWAS replication. The TWAS was performed using S-PrediXcan as described above. In GTEx lung, models were obtained for 11,518 genes. Bonferroni-corrected TWAS gene was thus set at $P_{TWAS} < 4.34E−6$. We also sought replication of TWAS genes identified in our lung eQTL dataset. Significant replication was considered for genes with the same direction of effect and with $P_{TWAS} < 0.05$ in GTEx lung.

**Bayesian colocalization**. For specific asthma-associated loci and genes, we evaluated whether the asthma GWAS and lung eQTL signals shared the same causal variants using the COLOC package in R[14]. For the loci of interest, summary statistics from the asthma GWAS in UK Biobank were combined with our lung eQTL results using a window of 1 Mb upstream and downstream of the TWAS genes. We considered colocalization events when the posterior probability of shared eQTL and GWAS associations (PP4) was greater than 60%. The colocalization analyses for the 485 blood eGenes were performed using the same method, but using the blood eQTL results from the eQTLGen Consortium (www.eqtlgen.org)[9]. LocusCompare[58] was used to visualize GWAS and eQTL colocalization events.

**TWAS in GTEx blood, skin, small intestine, and spleen**. S-PrediXcan was also used to explore TWAS genes in other asthma relevant tissues. GTEx v8 datasets for blood ($n = 670$), skin not sun exposed ($n = 517$), skin sun exposed ($n = 605$), small

intestine ($n = 174$), and spleen ($n = 227$) were analysed with the asthma GWAS. Significant gene expression models (FDR < 0.05) were obtained for 10,210 genes in blood, 12,347 genes in skin not sun exposed, 13,375 genes in skin sun exposed, 5184 genes in small intestine, and 8473 genes in spleen. Bonferroni-corrected TWAS genes were thus set at $P_{TWAS} < 4.90E−6$, $P_{TWAS} < 4.05E−6$, $P_{TWAS} < 3.74E−6$, $P_{TWAS} < 9.65E−6$, and $P_{TWAS} < 5.90E−6$, respectively.

**Cell type and tissue enrichment of asthma-associated loci**. We used GARFIELD[19] to overlap our GWAS findings with regulatory and functional annotations derived from ENCODE, GENCODE and Roadmap Epigenomics projects. A total of 1005 annotation features were considered including chromatin states, histone modifications, genic annotations, transcription factor binding sites, and open chromatin data (FAIRE, DNase I hypersensitivity site hotspots, peaks, and footprints), which were evaluated in different cell types and tissues. LD pruning of GWAS SNPs was performed at $r^2 > 0.8$ and fold enrichment was evaluated at two GWAS significance thresholds: 1.0E−5 and 1.0E−8.

**Functional mapping and annotation in blood**. We used the FUMA platform[17] to functionally annotate our GWAS findings. The summary statistics of the asthma GWAS in UK Biobank were uploaded in FUMA. The SNP2GENE function was used to map GWAS SNPs to 1) deleterious coding SNPs (positional mapping), 2) blood eQTL (eQTL mapping), and 3) chromatin contact interactions (chromatin interaction mapping). Positional mapping was performed by selecting exonic variants directly associated with asthma ($P_{GWAS} < 5E−8$) or in LD with asthma-associated variants using a LD $r^2$ threshold of 0.6 based on the 1000 Genomes EUR reference panel. Protein coding variants (excluding synonymous) with CADD score > 20 were further prioritized. SIFT and PolyPhen-2 scores were obtained from dbNSFP[59]. Blood *cis*-eQTL mapping was performed using a publicly available dataset of 31,684 samples[9]. Significant SNP-gene pairs ($P_{FDR} < 0.05$) were identified and then mapped to genetically expressed genes associated with asthma, or eGenes. Chromatin interaction mapping was performed using Hi-C data of a lymphoblastoid B cell line (GM12878, GEO accession number GSE87112). Results of eQTL and chromatin mapping were visualized using circos plots generated by FUMA.

**Mendelian randomization in blood with asthma**. Two-sample summary-level Mendelian randomization analyses were performed to infer causal associations between blood eGenes and asthma. The genetic effects on asthma risk were derived from the current GWAS in UK Biobank and the genetic effects on gene expression in blood were derived from a published eQTL dataset[9]. Mendelian Randomization was performed using the inverse-variance weighted (IVW) and Egger methods as implemented in the MendelianRandomization package in R. SNPs were selected within a window of 500 Kb around the transcription start site of each blood eGene. SNPs associated with gene expression ($P < 0.001$ corresponds to ~$F$ statistics > 10) and independent ($r^2 < 0.1$ based on the 1000 Genomes EUR reference panel) were selected as instrumental variables. We requested at least three instrumental variables per gene to perform Mendelian randomization. A $P$ value below the Bonferroni threshold was considered as significant (431 Mendelian Randomization with enough instrumental variables: $P_{Bonferroni} < 0.05/431 < 1.16E−4$). The Cochran's Q-test and MR-PRESSO (Mendelian randomization pleiotropy residual sum and outlier) global test were used to determine the presence of unmeasured pleiotropy. Heterogeneity ($P_{Q-test} < 0.05$) was corrected by applying the MR-PRESSO approach[60].

**Druggable target genes**. A list of druggable genes were obtained from the drug-gene interaction database[22] (DGIdb, www.dgidb.org) and the druggable genome[23]. Target genes of asthma-associated variants identified by TWAS, eQTL, and chromatin interactions were integrated and overlaid with the list of druggable genes. Druggable target genes were then queried for candidate drugs, interaction types and clinical indications in DGIdb[22], DrugBank (www.drugbank.ca), ChEMBL (www.ebi.ac.uk/chembl), and PubChem (pubchem.ncbi.nlm.nih.gov). Target genes were also queried on the Open Targets Platform[21] for their association with asthma.

**Phenome-wide association study (PheWAS)**. The potential effects of genes prioritized as therapeutic targets for asthma were evaluated using a PheWAS approach. The GWAS sentinel variants were queried in the GeneATLAS database[26], which contains genetic association results for 778 traits in European individuals from the UK Biobank. For sentinel variants not available in GeneATLAS, proxies were identified using LDlink[61]. Traits were considered significant using the default threshold of $P_{PheWAS} < 1E−8$ in GeneATLAS.

**Genome build**. GRCh37 (hg19) coordinates were used in this study.

**Reporting summary**. Further information on research design is available in the Nature Research Reporting Summary linked to this article.

## Data availability

The summary statistics for the asthma GWAS in UK Biobank ($n = 56,167$ asthma cases and 352,255 controls) are available at The NHGRI-EBI Catalog of human genome-wide association studies: https://www.ebi.ac.uk/gwas/, study accession GCST90014325. The human lung tissue eQTL study is available in dbGaP under accession phs001745.v1.p1. The full summary statistics for the lung asthma TWAS (19,918 probe sets with significant gene expression prediction models) are available in Supplementary Data 21. Summary statistics from the Trans-National Asthma Genetic Consortium were downloaded from the GWAS catalog: https://www.ebi.ac.uk/gwas/downloads/summary-statistics, study accession GCST006862.

## Code availability

The following software packages were used as part of this study: SAIGE, version 0.36.3.1: https://github.com/weizhouUMICH/SAIGE. LDSC, version 1.0.1: https://github.com/bulik/ldsc, GCTA, version 1.93.2beta: https://cnsgenomics.com/software/gcta/#COJO, S-PrediXcan: https://github.com/hakyimlab/MetaXcan, COLOC, version 3.2.1: https://cran.r-project.org/web/packages/coloc/index.html, LocusCompareR: version 1.0.0, https://github.com/boxiangliu/locuscomparer, GARFIELD: version 2, https://www.ebi.ac.uk/birney-srv/GARFIELD/, FUMA: http://fuma.ctglab.nl, Enrichr: https://maayanlab.cloud/Enrichr/, MendelianRandomization: https://cran.r-project.org/web/packages/MendelianRandomization, MR-PRESSO: https://github.com/rondolab/MR-PRESSO, LocusZoom: https://github.com/Geeketics/LocusZooms, UpSet plot: https://github.com/hms-dbmi/UpSetR, Chromosome ideogram: http://visualization.ritchielab.org/phenograms/document, GeneATLAS: http://geneatlas.roslin.ed.ac.uk, LDlink: https://ldlink.nci.nih.gov, dbNSFP: http://database.liulab.science/dbNSFP.

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

## Acknowledgements

The authors would like to thank the team at the IUCPQ-UL Biobank for their valuable assistance. This research has been conducted using the UK Biobank Resource. This study was supported by grants from the Chaire de pneumologie de la Fondation JD Bégin de l'Université Laval, the Fondation de l'Institut universitaire de cardiologie et de pneumologie de Québec, and the Canadian Institutes of Health Research (MOP—123369). Patrick Mathieu holds a Fonds de Recherche du Québec-Santé (FRQS) Research Chair. Sébastien Thériault holds a Junior 1 Clinical Research Scholar award from the FRQS. Yohan Bossé holds a Canada Research Chair in Genomics of Heart and Lung Diseases.

## Author contributions

K.V., P.M., S.T., and Y.B. contributed to the conception and study design. N.G., P.J., M.O., M.B., W.T., D.S., D.N., K.H., P.M., S.T., and Y.B. contributed to data collection. K.V., Z.L., V.B.-B., A.C., J.-C.B., A.E., J.L., P.M., S.T., and Y.B. contributed to data analysis. K.V., Z.L., V.B.-B., N.G., C.L., K.G., A.C., M.L., L.-P.B., P.M., S.T., and Y.B. contributed to data interpretation. K.V., P.M., S.T., and Y.B. drafted the manuscript. All authors revised the manuscript.

## Competing interests

The authors declare no competing interests.
