## [Transparent Peer Review File · Communications Biology]

Reviewers' comments:

Reviewer #1 (Remarks to the Author):

Summary

Valette et al. have attempted to identify causal genes for asthma with a view to identify potential therapeutic targets. They performed a GWAS of asthma in UK Biobank, then firstly looked for a functional effect of GWAS variants by looking for exonic variants that alter protein coding within each GWAS signal. Secondly, they combined the GWAS data from UK Biobank with expression data in external reference samples to infer genetically predicted expression to see if this was associated with asthma for genes expressed in lung and blood (TWAS). They used complimentary techniques to associate genetically driven expression with asthma including colocalisation and 2-sample Mendelian randomisation. They also looked for enrichment of asthma loci in regulatory regions and looked for enrichment in biological pathways of the genes whose expression in blood was associated with asthma.

The main results presented are 9 previously unreported GWAS asthma loci. 8 genes are implicated by deleterious coding variants outside the MHC. 55 genes are implicated by lung TWAS across 21 asthma loci. 485 genes are implicated by look up in blood eQTL and 563 genes are implicated by chromatin contact mapping. Finally, a list of 21 genes are prioritized as therapeutic targets for asthma based on the criteria that (i) the gene is implicated by 2 of the 3 methods, (ii) are listed as associated with asthma in Open Targets and (iii) are druggable in either the DGIdb or the druggable genome.

Overall impression

The paper is well written, clearly presented and the methods seem to have been implemented appropriately. I think there is an important contribution to the understanding of the genetics of asthma in this work although the authors

do not emphasise sufficiently which are the novel results that add new understanding and which are replicating association of known asthma genes perhaps via a different methodology; also important results nevertheless.

As TWAS results have been shown to be ambiguous in identify causal genes it is good that the authors used additional methods to establish a causal link such as colocalisation and Mendelian randomisation, although I am unclear as to why certain methods were applied to certain expression data sets instead of used across the board.

I am pleased to see the depth of information provided in the Supplementary tables and it appears there is enough information provided to repeat this analysis given access to the relevant UK Biobank fields and the lung and blood

eQTL resources. I hope full summary statistics of the GWAS and TWAS will be made available on publication.

The authors show the relevance of this work for identifying candidate therapeutic targets.

Major comments

1. For the 9 novel GWAS signals are there any publicly available genome-wide summary statistics available in order to look for replication (either of the sentinel or proxies) in other cohorts? e.g. TAGC consortium (PMID: 29083406).

2. For the 116 independent signals in Supplementary Table 2, how many are novel? i.e. in addition to reporting 9 novel loci, are you reporting any novel signals in previously reported loci that are independent from the previously reported signals in those loci?

3. How many of the previously reported signals replicated in your data set? You replicate association for 64 but were there any previously reported that did not come up as significant in your results – this is important if this is (one of) the largest GWAS of asthma to date? Perhaps include an additional

supplementary table with all previously reported signals comparing the previous OR and P with the values in this study – or best proxy if original sentinel is not present.

4. Need to be explicit about the rationale behind the alternative study designs e.g. what's the relevance of excluding controls with atopy? And why did you choose the study design for the main analysis over these.

5. Loose criteria for "Positional mapping of deleterious coding SNPs"; I would call it "Functional annotation of coding SNPs" as "positional mapping" implies you did something more elaborate such as using a finemapping method (see minor comment about overuse of the term "map"). Criteria to associate exonic variants with asthma: $r^2 \geq 0.6$ appears to mean in LD with any variant with $P < 5 \times 10^{-8}$ as there are variants in column K in supplementary table 3 that are not in supplementary table 2? Should really be r^2 relative to the 116 independent signal sentinels, or ideally in a credible set calculated around the sentinels, otherwise with this criteria the exonic variant could be outside the asthma signal if it is only in LD with a variant in the tail region (albeit still $P < 10^{-8}$) of an asthma signal. Why use 1000G EUR for the LD calculation instead of the UK Biobank data itself; which was used for LD in the conditional analysis?

6. Are any of the genes in Table 2 novel for asthma? Line 135: "Genes of known biological relevance were identified including filaggrin"; line 137: "three potential target genes were identified", reads like they are novel findings, if they have been previously reported for asthma then should say "deleterious variants within previously reported gene". Are any of the deleterious variants are being reported for the first time?

7. Again, for Table 3, which are newly identified asthma genes? Also highlight on Figure 3 and Supplementary Figure 4.

8. The replication p value for GTEx should be Bonferroni corrected for the number of genes looked up not $P < 0.05$. Also, I'm not sure it counts as TWAS replication unless you have an independent GWAS as well as an independent eQTL.

9. I can't see why different methodologies were used for different eQTL data sets:

- Why did you do a TWAS with the lung data and a cis-eQTL look up in the blood data? Why not do both in both?
- Why was coloc used for the lung data and HyPrColoc for the blood data?
- Why was Enrichr used for the blood data but not the lung data?
- Why was 2 sample MR used for blood and not lung data?

10. Again, via blood eQTL, which are the newly identified asthma genes? Highlight in Supplementary Table 8?

11. GARFIELD showed enrichment in other cell lines too, why did you only do chromatin contact mapping for GM12878?

12. Figure 5 has lots of information, what's the takeaway message from it?

13. Not convinced that genes that overlap 2 of the 3 methods is a good criterion. Might not a gene whose expression is only associated in lung but not in blood, or vice-versa, be an important target nevertheless? If you went with criteria 2 & 3, you would prioritise 44 genes, adding some of those that are already targets of investigational or approved asthma drugs: CCR4, IL13, IL2RA, IL5.

Minor comments

1. Not keen on widespread use of the word "map" e.g. line 73: "The objective of this study was to

map candidate causal genes of asthma in lung and blood tissues". I'd say "identify candidate causal genes" because as written it is not clear what is being mapped from and to, SNP associations to genes? Likewise, lines 126, 199, 262 could all be replaced with "identify"; correct use on line 536 though.

2. Region plots more readable as 2 per column and include significance threshold.

3. Line 100 "Nevertheless, we discarded this locus as more validation is needed to robustly establish its association with asthma", this could be said about all loci, especially the novel ones, seems to imply you've robustly established the association in other loci, how? (see major comment about looking for replication).

4. Line 120, say "discrepancy" rather than "outlier", it's not really a case of being an outlier in the usual sense of being an unexpected extreme value. The sensitivity analysis is specifically looking for differences.

5. Supplementary Table 3: why only CADD scores? You mention SIFT and Polyphen in the discussion, would be useful to include these too.

6. Supplementary Table 3: what are columns Q-V? All supplementary tables need a key describing what all the columns are.

7. Line 144: reference for the lung eQTL data set?

8. Table 3: would be helpful to list the sentinel SNP from the GWAS that the TWAS gene pertains to rather than just the "Chr band". Perhaps also highlight those genes that colocalised and highlight where directions of effect agreed instead of the reader having to cross-check.

9. Line 170: "21 asthma-associated loci", there are 22 in Table 3.

10. Supplementary Figures 4 & 6: what are the unlabelled blobs?

11. The correction by MR-PRESSO didn't change the number of causally associated genes? Should make this explicit.

12. Line 199: In reference 11 they say that trans-eQTLs are more informative than cis-eQTLs, did you consider looking at trans-eQTLs?

13. Line 266: "As expected, the yield of deleterious coding variants was low", add number in brackets.

14. Lines 313-322: I don't think this general review on the state of GWAS adds anything to the discussion of these results.

15. Line 342: "have identified 5 independent significant variants", which did you identify and which were already reported?

16. Line 391: "we have not taken into account likely modifiers of genetic risk", could you have checked for a smoking interaction effect?

17. Could a PheWAS look up of prioritised genes give you a heads up about any potential side effects of targeting these genes?

18. Line 468: mention the option "--cojo-slct" without specifying which software this pertains to.

19. For GCTA –cojo-slct which LD reference panel was used? Presumably the same genotypes as the GWAS?

20. Supplementary Table 14 should be referenced at line 409 in the Methods.

21. Line 538: the details of the positional mapping of asthma GWAS variants should be higher up so that methods are in the same order as results.

22. Some references in Supplementary Table 1 are not in the list of references e.g. Zhu_2019_EurRespirJ

Reviewer #2 (Remarks to the Author):

This is an interesting and timely study providing an advancement in our understanding of asthma including the identification of genetic variants associated with disease risk but also importantly leveraging different approaches to identify the genes underpinning these observations. This provides new understanding but also new targets for drug development and repositioning. Overall the manuscript is divided into five main sections; i) GWAS of asthma in UKB, ii) TWAS using two lung tissue samples, iii) identification of eGenes in blood, iv) functional annotation of variants and v) evaluation of gene targets as drug opportunities. Overall, this approach works well with ultimately the authors highlighting 21 genes/proteins that warrant prioritisation in existing/new drug development for asthma. While an addition to the field several issues need resolving prior to potential publication.

Major comments

1. The GWAS in UKB using 56k cases and 352k controls is the largest to date mainly by leveraging some additional cases from records, could the authors clarify this and how this study compares to the now many studies using UKB for GWAS of asthma? In particular how different to that reported by Johansson et al 2019, PMID: 31361310. In the Johansson study 41k asthma cases were included with 239k controls. This study is not included in the current manuscript? In addition a quick look suggest that the Johansson study has previously reported several of the “novel” signals included in the current manuscript, e.g. ADAMTSL3.

2. The authors report 73 loci with 116 independent signals in the GWAS and highlight nine potential novel signals (with the caveat outlined in 1). It is important to note that no replication is attempted for these potentially novel signals which is standard practice when reporting new genetic findings. While the same sample size is unlikely to be achieved, the authors should investigate these signals in additional cohorts to provide robustness even at nominal significance, e.g. 23&Me, GABRIEL cohorts.

3. In the GWAS, a statistical threshold of 5×10^{-8} is used which is historical when GWAS was conceived and prior to large scale imputation, can the authors justify this when now testing >35 million variants?

4. The sensitivity analyses of the genetic association testing is well conceived to investigate other lung diseases, smoking and allergy. For any novel signals (see points 1 and 2 above) can the authors include a summary of the findings from the sensitivity analyses for the sentinel SNPs.

5. The TWAS approach using two sets of lung samples is a strength and a limitation, more details/discussion should be included regarding the subjects that donated material and the potential heterogeneity with the lung eQTL dataset, e.g. cell composition/disease status (how many donors of lung samples had asthma?, COPD?) etc.

6. The identification of deleterious variants in LD with the signals is useful, in addition to the current focus on p-values, could the authors specifically comment on effect sizes for these variants particularly when some striking e.g. stop codon in FLG.

7. IF I understand correctly, 24/55 genes identified in the Lung eQTL TWAS could not be evaluated in the replication GTEx dataset as expression data was not available. This is a major limitation and should be highlighted/discussed.

8. ENCODE/Roadmap Epigenomics Project data is used to help investigate the link between functional marks/variants and genes. This is useful, however these datasets have limited respiratory relevant cells and so this should be acknowledged as a limitation.

9. For the blood based analyses involving 31,684 samples, why was a discovery/replication approach not used to provide confidence in findings?

10. The current manuscript would greatly benefit from a direct comparison of the main findings of the current study and discussion of a similar paper that used published GWAS data, eQTL datasets and drug information to identify priority candidates for drug development in asthma see PMID: 32910899.

Minor comments

1. Throughout "top SNPs" are referred to, it is unclear how this is defined.

2. Table 3 would benefit from being focussed to replicated findings in both lung samples with all data put in Suppl.

Reviewer #3 (Remarks to the Author):

In this manuscript, the authors leveraged lung and blood transcriptome as well as epigenetic marks to prioritize the most likely causal genes within asthma susceptibility loci derived from the UK Biobank. The authors should be commended in their use of various bioinformatics approaches. I have a couple of comments:

1. There is no functional validation to investigate the biological role of the candidate causal genes in the pathogenesis of asthma. Future follow-up studies are needed to demonstrate causality in order to address the clinical needs of asthma patients. It would be good if the authors could provide more evidence on biological relevance and clinical implications of the prioritized candidate causal genes and the added value of your findings.

2. Asthma is a clinically heterogeneous disease, and distinct loci have been reported underlying the susceptibility to childhood-onset asthma and adult-onset asthma (PMID: 31036433). Previous gene expression and tissue enrichment patterns suggest that genes at the childhood onset loci were most highly expressed in skin, blood and small intestine, whereas genes at the adult onset loci were most highly expressed in lung, blood, small intestine and spleen. Therefore, the role of age of asthma onset should be taken into account in GWAS to prioritize candidate causal genes, and TWAS analyses should also be conducted in the other asthma relevant tissue such as skin, small intestine and spleen that are available in GTEx consortium.

3. What is the age range of samples in the eQTL datasets used in this analysis? If the samples were mainly from adults, this study may miss relevant genes whose expression is environmentally regulated.

4. The authors did not provide any replication of the asthma loci identified in the UK Biobank. The lack of a replication data set should be listed as a limitation.

5. GARFIELD analysis showed strong enrichment in a variety of cell types in blood including GM12878

cell line, CD19_Primary_Cells, CD4_Primary_Cells and CD8_Primary_Cells, it would be helpful to additionally use Hi-C data obtained in other cell types in blood for chromatin contact mappings.

6. What is the possible reason that lung is not significant for enrichment of DNase I hypersensitive sites?

7. It has been suggested that, for validating drug targets, the gene must include multiple causative variants of known biological function (PMID: 23868113). As some asthma loci identified in this study contain independent variants that are not in LD, these SNPs could be used to investigate dose-response curves. It would be more convincing evidence if the genes prioritized as therapeutic targets for asthma could demonstrate a dose-response relationship.

Responses to reviewers

We would like to thank the reviewers for taking the time to evaluate our manuscript. You will find below a point-by-point response to comments. The comments from the reviewers are provided verbatim in bold.

Reviewer #1:

Summary

Valette et al. have attempted to identify causal genes for asthma with a view to identify potential therapeutic targets. They performed a GWAS of asthma in UK Biobank, then firstly looked for a functional effect of GWAS variants by looking for exonic variants that alter protein coding within each GWAS signal. Secondly, they combined the GWAS data from UK Biobank with expression data in external reference samples to infer genetically predicted expression to see if this was associated with asthma for genes expressed in lung and blood (TWAS). They used complimentary techniques to associate genetically driven expression with asthma including colocalisation and 2-sample Mendelian randomisation. They also looked for enrichment of asthma loci in regulatory regions and looked for enrichment in biological pathways of the genes whose expression in blood was associated with asthma.

The main results presented are 9 previously unreported GWAS asthma loci. 8 genes are implicated by deleterious coding variants outside the MHC. 55 genes are implicated by lung TWAS across 21 asthma loci. 485 genes are implicated by look up in blood eQTL and 563 genes are implicated by chromatin contact mapping. Finally, a list of 21 genes are prioritized as therapeutic targets for asthma based on the criteria that (i) the gene is implicated by 2 of the 3 methods, (ii) are listed as associated with asthma in Open Targets and (iii) are druggable in either the DGIdb or the druggable genome.

Overall impression

The paper is well written, clearly presented and the methods seem to have been implemented appropriately. I think there is an important contribution to the understanding of the genetics of asthma in this work although the authors do not emphasise sufficiently which are the novel results that add new understanding and which are replicating association of known asthma genes perhaps via a different methodology; also important results nevertheless.

As TWAS results have been shown to be ambiguous in identify causal genes it is good that the authors used additional methods to establish a causal link such as colocalisation and Mendelian randomisation, although I am unclear as to why certain methods were applied to certain expression data sets instead of used across the board.

I am pleased to see the depth of information provided in the Supplementary tables and it appears there is enough information provided to repeat this analysis given access to the relevant UK Biobank fields and the lung and blood eQTL resources. I hope full summary statistics of the GWAS and TWAS will be made available on publication.

Summary statistics were added in the revised manuscript. We also included these sentences in the Data availability section:

The summary statistics for the asthma GWAS in UK Biobank (n=56,167 asthma cases and 352,255 controls) are available at The NHGRI-EBI Catalog of human genome-wide association studies: <https://www.ebi.ac.uk/gwas/>. The full summary statistics for the lung asthma TWAS (19,918 probe sets with significant gene expression prediction models) are available in the Supplementary Data 1.

The authors show the relevance of this work for identifying candidate therapeutic targets.

Major comments

1. For the 9 novel GWAS signals are there any publicly available genome-wide summary statistics available in order to look for replication (either of the sentinel or proxies) in other cohorts? e.g. TAGC consortium (PMID: 29083406).

We have looked at the summary statistics from the Trans-National Asthma Genetic Consortium (Demenais et al. Nat Genet 2018), which is the largest asthma GWAS without UK Biobank.

We first looked for the sentinel genetic variants at each of the 9 novel loci. For 8 loci the sentinel variants were not available in TAGC, which was expected as this GWAS was based on HapMap2-imputed data. The next most significant variant in our study at each locus was then evaluated until an overlapping variant with TAGC was found. The results are provided in this table.

Loci	Chr	SNP ^a	LD (r ²) ^b	UKB		TAGC	
				beta	p	beta	p
9	2	rs567095143	0.707	-0.037	4.54E-08	NA	NA
		rs11677053		-0.0360	7.18E-08	-0.015194	0.25202
26	6	rs533610689	0.671	-0.039	5.87E-09	NA	NA
		rs12523702		-0.0366	6.56E-08	-0.004819	0.72446
30	6	rs7770794		0.042	8.81E-09	-0.0088	0.5437
31	7	rs112119265	0.317	-0.078	1.76E-08	NA	NA
		rs7809448		-0.0378	1.95E-05	-0.000733	0.96678
35	7	rs576468798	0.0002	2.012	2.00E-08	NA	NA
		rs740214		0.0302	2.53E-03	0.021040	0.27376
38	8	8:134185116_TA_T	0.997	-0.040	3.88E-08	NA	NA
		rs4736638		-0.0397	4.10E-08	-0.006606	0.63660
44	10	rs1870140	0.994	-0.0498	3.96E-08	NA	NA
		rs6586030		-0.0478	1.23E-07	-0.008127	0.63852
60	15	rs4842921	0.612	-0.041	1.17E-09	NA	NA
		rs2401171		-0.0384	4.01E-09	-0.036477	0.00341
68	17	rs112267124	0.992	0.043	2.55E-08	NA	NA
		rs17581728		0.0427	2.83E-08	0.016164	0.28080

^aFor each locus, the sentinel variant is listed first. For sentinel variants not available in TAGC, the next most significant variant overlapping with TAGC is provided.

^bLD from individuals of white British ancestry in UK Biobank.

Highlighted in yellow are loci that are no longer considered novel. See comment 1 of reviewer #2: rs112119265-CARD11 on chromosome 7 and rs4842921-ADAMTSL3 on chromosome 15 were identified by Johansson et al. Hum Mol Genet 2019.

This was added in the Results section:

We checked for potential replication for the novel loci in summary statistics from the Trans-National Asthma Genetic Consortium (TAGC) comparing 19,954 European ancestry cases and 107,715 European ancestry controls. The sentinel variants or the next most significant variants overlapping with TAGC were not associated with asthma ($P_{\text{GWAS}} > 0.05$).

This was added in the Discussion section:

We did not observe replication of these novel loci in the Trans-National Asthma Genetic Consortium (Demenais et al. Nat Genet 2018, PMID 29273806) and they should thus be interpreted with caution.

This was added in the Data availability section:

Summary statistics from the Trans-National Asthma Genetic Consortium were downloaded from the GWAS catalog: <https://www.ebi.ac.uk/gwas/downloads/summary-statistics>, study accession GCST006862.

2. For the 116 independent signals in Supplementary Table 2, how many are novel? i.e. in addition to reporting 9 novel loci, are you reporting any novel signals in previously reported loci that are independent from the previously reported signals in those loci?

This concerns 16 loci with more than one independent signals. For each of these loci, we have evaluated the LD (r^2 , white British ancestry from UKB) between our independent variants and those reported in previous asthma GWAS. The results are in Supplementary Table 3.

This sentence was added in the Results section:

Two independent signals at the MHC locus (rs2517761 and rs2523430) and two at the 1q21-*FLG* locus (rs185433896 and rs558312428) were independent ($r^2 < 0.1$) from asthma-associated variants reported in the literature (Supplementary Table 3).

3. How many of the previously reported signals replicated in your data set? You replicate association for 64 but were there any previously reported that did not come up as significant in your results – this is important if this is (one of) the largest GWAS of asthma to date? Perhaps include an additional supplementary table with all previously reported signals comparing the previous OR and P with the values in this study – or best proxy if original sentinel is not present.

For the first question, the header “Previous_asthma_GWAS” in Supplementary Table 1 shows all previous asthma GWAS that have reported each locus.

We added Supplementary Table 4 that includes previous signals not significantly associated with asthma in our study.

This was added in the Results section:

Genetic association results for previous asthma GWAS signals that are not significant in this study are provided in Supplementary Table 4.

4. Need to be explicit about the rationale behind the alternative study designs e.g. what’s the relevance of excluding controls with atopy? And why did you choose the study design for the main analysis over these.

We have modified (in italic here) this paragraph in the Results section:

Three alternative study designs (2, 3 and 4) were evaluated to investigate the potential confounding effects of other lung diseases, smoking and allergy. The aforementioned results are considered study design 1 and the main analysis. *This design was selected to maximize sample size and statistical power.* Study design 2 excluded cases and controls with COPD, emphysema, chronic bronchitis, interstitial lung disease or alpha-1 antitrypsin deficiency (n=47,391 cases and 340,033 controls), *because similarities in their clinical presentation can result in misclassification of cases and controls.* Study

design 3 excluded cases and controls with a positive smoking history (n=21,097 cases and 136,586 controls). *This analysis was done to further evaluate the potential confounding effects of smoking-related lung disease, most particularly COPD, on the asthma case-control status.* Study design 4 excluded controls with atopy including hay fever, allergic rhinitis, and eczema/atopic dermatitis (n=56,167 cases and 268,142 controls). *This study design explored the impact of excluding from the control group individuals who suffer from other genetically correlated allergic diseases, which may help to delineate unique vs. shared genetic etiology of asthma and allergy.* Case-control genetic association analyses were thus performed on these three alternative study designs.

5. Loose criteria for “Positional mapping of deleterious coding SNPs”; I would call it “Functional annotation of coding SNPs” as “positional mapping” implies you did something more elaborate such as using a finemapping method (see minor comment about overuse of the term “map”). Criteria to associate exonic variants with asthma: $r^2 \geq 0.6$ appears to mean in LD with any variant with $P < 5 \times 10^{-8}$ as there are variants in column K in supplementary table 3 that are not in supplementary table 2? Should really be r^2 relative to the 116 independent signal sentinels, or ideally in a credible set calculated around the sentinels, otherwise with this criteria the exonic variant could be outside the asthma signal if it is only in LD with a variant in the tail region (albeit still $P < 10^{-8}$) of an asthma signal. Why use 1000G EUR for the LD calculation instead of the UK Biobank data itself; which was used for LD in the conditional analysis?
We changed the subtitle for “Functional annotation of coding SNPs”.

Note that the conditional analysis was performed with COJO (Yang et al. Nat Genet 2012, and results are in Supplementary Table 2), while coding variants analysis was performed using FUMA (Watanabe K et al. Nat Commun 2017, with results in Supplementary Table 6). We believe this is clear from the Methods section, but the reviewer raised a good point, this is not clear from the Results section. We have thus clarified the Results section:

Our first strategy to prioritize target genes within GWAS-nominated asthma loci was to identify deleterious coding variants. This step was performed in FUMA (Watanabe K et al. Nat Commun 2017). FUMA takes GWAS summary statistics and derives its own list of independent significant variants ($P_{\text{GWAS}} < 5.0 \times 10^{-8}$ and $r^2 < 0.6$ based on 1000G EUR). All known proxies that have $r^2 \geq 0.6$ with one of the independent significant variants, regardless of being available in the GWAS input file, are then considered candidate variants and included for further annotation. Supplementary Table 6 shows 354 exonic candidate variants identified by this method. These variants are located in 27 loci and two-thirds (236 out of 354) of them are in the MHC locus.

6. Are any of the genes in Table 2 novel for asthma? Line 135: “Genes of known biological relevance were identified including filaggrin”; line 137: “three potential target genes were identified”, reads like they are novel findings, if they have been previously reported for asthma then should say “deleterious variants within previously reported gene”. Are any of the deleterious variants are being reported for the first time?

We added this sentence in the Results section:

All genes in Table 2 have been reported in previous asthma GWAS.

7. Again, for Table 3, which are newly identified asthma genes? Also highlight on Figure 3 and Supplementary Figure 4.

We added this in the Results section:

Among the 55 lung TWAS genes, nine were not reported in previous asthma GWAS. This includes *CSF2* and *HSPA4* on 5q31, *TRIM10* on 6p22-p21, *C9orf38* on 9p24, *TSPAN14* on 10q23, *FAM62A* on 12q13, *MAP2K5* on 15q22-q23, *CRKRS* and *PERLD1* on 17q12-q21.2.

These nine genes are now underlined in Table 3 and in blue in Figure 3 and Supplementary Figure 5.

8. The replication p value for GTEx should be Bonferroni corrected for the number of genes looked up not $P < 0.05$. Also, I'm not sure it counts as TWAS replication unless you have an independent GWAS as well as an independent eQTL.

In a TWAS, we used an eQTL dataset to impute the cis genetic component of expression into a larger set of cases and controls using their SNP genotype data. The approach can be conceptualized as having imputed expression data for all cases and controls used in the GWAS without directly measuring expression levels in these samples. It is important to understand that we can only impute the genetic component of expression and this part of expression that can be explained by SNPs is dependent on the eQTL dataset only. So, since we used an independent eQTL set, this constitutes a replication even if we used the same GWAS dataset. We agree with the reviewer that an independent GWAS would be even more robust. However, an adequately powered replication set seems impractical now to match our UK Biobank GWAS of 56,167 asthma cases and 352,255 controls. Note that we have used this lung TWAS replication strategy before as part of large consortia (PMID 31696517 and 29547942).

There is no standard in terms of the most appropriate TWAS replication P value threshold. As indicated in the Methods section: "Significant replication was considered for genes with the same direction of effect and with $P_{\text{TWAS}} < 0.05$ in GTEx lung." We believe this is reasonable knowing the differences that exist between our lung eQTL study and GTEx lung in terms of sample size (1,038 vs. 515), human lung sampling (surgery vs. post-mortem), tissue processing (methods for freezing, storing, thawing tissues and extracting RNA), and gene expression platforms (microarray vs. RNA-Seq). Other investigators have also raised concerns about the lung transcriptome data in GTEx (see PMID 27588449).

All these considerations aside, Table 3 provides the exact P values for the lung TWAS genes in GTEx. The readers can thus judge the robustness of the replication.

See also our response to comment #7 of reviewer #2.

9. I can't see why different methodologies were used for different eQTL data sets:

• Why did you do a TWAS with the lung data and a cis-eQTL look up in the blood data? Why not do both in both?

We cannot do a TWAS with the blood data. The blood eQTL dataset used in this study is from the eQTLGen Consortium (Võsa et al. biorxiv 2018). The main advantage of this dataset is the sample size ($n=31,684$), but we only have access to summary statistics. We need individual genotyping data and gene expression data (or precomputed gene expression weights) from the eQTL dataset to perform a TWAS.

Note that based on this comment as well as comment 2 of reviewer #3, we added blood TWAS from GTEx in the revised manuscript. The blood eQTL dataset from GTEx is based on a smaller sample size ($n=670$), so we prefer to keep eQTLGen as the main source of data for blood analyses.

• **Why was coloc used for the lung data and HyPrColoc for the blood data?**

We are only using COLOC in the revised manuscript. We have made the corresponding changes in Supplementary Tables 15 and 16 as well as the Methods and Results sections.

• **Why was Enrichr used for the blood data but not the lung data?**

We added pathway analysis for lung data in the revised manuscript. Full results are in Supplementary Table 8.

We added this in the Results:

The lung TWAS genes associated with asthma were enriched in the Kyoto Encyclopedia of Genes and Genomes (KEGG) for asthma ($P_{\text{adjusted}}=0.003$), antigen processing and presentation ($P_{\text{adjusted}}=0.003$) and Th17 cell differentiation ($P_{\text{adjusted}}=0.004$) (Supplementary Table 8).

• **Why was 2 sample MR used for blood and not lung data?**

For lung data, we have favored TWAS followed by further filtering with TWAS replication in GTEx lung and colocalization. In blood, we had to find alternative methods to find the best possible causal genes of asthma because, as explained above, we do not have raw genotyping and gene expression data. We have thus favored cis-eQTL look up with further filtering by MR and colocalization.

10. Again, via blood eQTL, which are the newly identified asthma genes? Highlight in Supplementary Table 8?

Blood eGenes associated with asthma are now reported in Supplementary Table 12. We added a column entitled Reported_previous_asthma_GWAS, which indicates if the gene was reported in previous asthma GWAS.

11. GARFIELD showed enrichment in other cell lines too, why did you only do chromatin contact mapping for GM12878?

The top cell line showing enrichment in GARFIELD is GM12878, which is a lymphoblastoid B cell line. Hi-C data for GM12878 is publicly available:

<https://www.ncbi.nlm.nih.gov/geo/query/acc.cgi?acc=GSE87112>

Adding Hi-C data from other cell lines would be feasible, but we start to feel that there are a lot of data in the revised manuscript (7 Figures, 6 Tables, 8 Suppl. Figures, and 20 Supplementary Tables). So we would prefer to keep the focus on the cell line showing the most enrichment in GARFIELD.

12. Figure 5 has lots of information, what's the takeaway message from it?

We want to illustrate candidate causal genes identified at the 17q12-q21 locus by our integrative genomic approach. To clarify this point, we have zoomed in at the 17q12-q21 locus, instead of showing the full chromosome 17. See the new Figure 6 in the revised manuscript.

We also added this in the Results:

Although interesting, prioritizing candidate genes for asthma was still challenging at many loci. For example, **Fig. 6** shows a zoom in of a circos plot for genetic association data integrating eQTL and chromatin contact mappings in GM12878 at the 17q12-q21 locus. Many eQTL genes (in green), Hi-C genes (in orange) or genes significant in both eQTL and Hi-C data (in red) were identified. We have thus decided to perform additional filtering using Mendelian Randomization (MR).

13. Not convinced that genes that overlap 2 of the 3 methods is a good criterion. Might not a gene whose expression is only associated in lung but not in blood, or vice-versa, be an important target nevertheless? If you went with criteria 2 & 3, you would prioritise 44 genes, adding some of those that are already targets of investigational or approved asthma drugs: CCR4, IL13, IL2RA, IL5. This is a good point. Thank you we have made the change in the revised manuscript. By excluding HLA molecules, we are now presenting 40 genes in Table 6.

Minor comments

1. Not keen on widespread use of the word “map” e.g. line 73: “The objective of this study was to map candidate causal genes of asthma in lung and blood tissues”. I’d say “identify candidate causal genes” because as written it is not clear what is being mapped from and to, SNP associations to genes? Likewise, lines 126, 199, 262 could all be replaced with “identify”; correct use on line 536 though.

Done

2. Region plots more readable as 2 per column and include significance threshold.

Done

3. Line 100 “Nevertheless, we discarded this locus as more validation is needed to robustly establish its association with asthma”, this could be said about all loci, especially the novel ones, seems to imply you’ve robustly established the association in other loci, how? (see major comment about looking for replication).

The regional plot for the 7p14 locus is now in Supplementary Fig. 3. We are concerned about this locus because there is only one rare variant with $P_{\text{GWAS}} < 5 \times 10^{-8}$ and no other variant that is even close to this threshold. This is not the case for the other loci (see Figure 2).

As indicated above, we clearly indicated that the novel GWAS loci are not replicated and will require further validation.

4. Line 120, say “discrepancy” rather than “outlier”, it’s not really a case of being an outlier in the usual sense of being an unexpected extreme value. The sensitivity analysis is specifically looking for differences.

Done

5. Supplementary Table 3: why only CADD scores? You mention SIFT and Polyphen in the discussion, would be useful to include these too.

We added the SIFT and Polyphen scores for nonsynonymous SNVs in Supplementary Table 6.

6. Supplementary Table 3: what are columns Q-V? All supplementary tables need a key describing what all the columns are.

For Supplementary Table 3 (that is now Supplementary Table 6), we have originally kept the standard output from FUMA. We have now added a description within the column headers or simply removed columns that were not relevant.

7. Line 144: reference for the lung eQTL data set?

Done

8. Table 3: would be helpful to list the sentinel SNP from the GWAS that the TWAS gene pertains to rather than just the “Chr band”. Perhaps also highlight those genes that colocalised and highlight where directions of effect agreed instead of the reader having to cross-check.

We added the sentinel variant from the GWAS in Table 3. Lung TWAS genes that replicated in GTEx lung are now in bold in Table 3.

9. Line 170: “21 asthma-associated loci”, there are 22 in Table 3.

We have recounted and there are 21.

10. Supplementary Figures 4 & 6: what are the unlabelled blobs?

These are now Supplementary Figures 5 and 8. We added this in the legends:

Unlabeled blobs are asthma GWAS loci for which no significant lung TWAS gene (or blood eGene) was identified.

11. The correction by MR-PRESSO didn’t change the number of causally associated genes? Should make this explicit.

We modified as follow:

Among the 50 blood eGenes, 26 did not show heterogeneity on the Cochrane’s Q test ($P > 0.05$), whereas 24 eGenes had heterogeneity ($P_{Q-test} < 0.05$). These 24 eGenes were corrected by applying the MR-PRESSO approach (methods) and all remain significant (Supplementary Table 16).

12. Line 199: In reference 11 they say that trans-eQTLs are more informative than cis-eQTLs, did you consider looking at trans-eQTLs?

This is interesting, but should be the focus of a different manuscript.

13. Line 266: “As expected, the yield of deleterious coding variants was low”, add number in brackets.

We modified as follow:

As expected, the yield of deleterious coding variants was low (8 nonsynonymous variants and 1 stop-gain variant with CADD score > 20), ...

14. Lines 313-322: I don’t think this general review on the state of GWAS adds anything to the discussion of these results.

We have removed that section.

15. Line 342: “have identified 5 independent significant variants”, which did you identify and which were already reported?

We have modified as follow:

FLG is located on chromosome 1q21.3, a locus where we have identified 5 independent significant variants, which have all been previously reported (Supplementary Tables 2 and 3).

16. Line 391: “we have not taken into account likely modifiers of genetic risk”, could you have checked for a smoking interaction effect?

We have addressed smoking with our sensitivity analysis.

17. Could a PheWAS look up of prioritised genes give you a heads up about any potential side effects of targeting these genes?

We added this in the Methods section:

Phenome-wide association study (PheWAS)

The potential effects of genes prioritized as therapeutic targets for asthma were evaluated using a PheWAS approach. The GWAS sentinel variants were queried in the GeneATLAS database (Canela-Xandri et al. Nat Genet 2018, PMID: 30349118), which contains genetic association results for 778 traits in European individuals from the UK Biobank. For sentinel variants not available in GeneATLAS, proxies were identified using LDlink (Machiela & Chanock. Bioinformatics 2015, PMID: 26139635). Traits were considered significant using the default threshold of $P_{\text{PheWAS}} < 1E-8$ in GeneATLAS.

We added this in the Results section:

Finally, we performed a cross-phenotype search in GeneATLAS (Canela-Xandri et al. Nat Genet 2018, PMID: 30349118) to evaluate potential effects of modulating genes prioritized as therapeutic targets for asthma. Diseases/traits from UK Biobank participants that are significantly associated with the GWAS sentinel variants underpinning the 40 prioritized target genes are indicated in Supplementary Table 18. Based on this *in silico* approach and available data, the safety profile of most of these targets seems favorable with genetic associations observed mostly with asthma-related phenotypes, white blood cells (eosinophil, neutrophil, lymphocyte), and other allergic conditions (hay fever/allergic rhinitis and allergy/hypersensitivity/anaphylaxis). Targets located in the MHC locus (*NOTCH4* and *ITPR3*) were associated with 41 different phenotypes, including ulcerative colitis and multiple sclerosis signals in the opposite direction from asthma. Similarly, the lead variant at the *IL5/IL13* locus linked with a decreased risk of asthma was associated with an increased risk of psoriasis.

We added this in the Discussion section:

We performed a quick PheWAS lookup on prioritized genes to provide a first appreciation of any potential side effects of targeting these genes. Although a more in-depth investigation will be needed on each gene, the direction of effect on asthma compared to other health conditions and most particularly allergic phenotypes seem to be concordant. Exceptions included genes located at the MHC locus showing a potential increase in risk of ulcerative colitis and multiple sclerosis and at the *IL5/IL13* locus showing a potential increase in risk of psoriasis.

This was added in the code availability section:
 GeneATLAS, <http://geneatlas.roslin.ed.ac.uk>
 LDlink: <https://ldlink.nci.nih.gov>

18. Line 468: mention the option "--cojo-slct" without specifying which software this pertains to.
 We changed for "GCTA --cojo-slct".

19. For GCTA --cojo-slct which LD reference panel was used? Presumably the same genotypes as the GWAS?

Exactly. We have modified as follow:

This was performed using a stepwise conditional analysis (GCTA --cojo-slct) (Yang et al. Nat Genet 2012) using UK Biobank as the LD reference panel.

20. Supplementary Table 14 should be referenced at line 409 in the Methods.

Done. This is now Supplementary Table 19.

21. Line 538: the details of the positional mapping of asthma GWAS variants should be higher up so that methods are in the same order as results.

OK we have reordered the methods in order to follow the results section as much as possible. However, positional mapping of asthma GWAS variants was performed in FUMA. We believe this is clearer now with changes made based on comment #5. However, FUMA was also used for blood eQTL mapping and chromatin interaction mapping. We think that it is simpler to describe FUMA in a single paragraph of the method section.

22. Some references in Supplementary Table 1 are not in the list of references e.g.

Zhu_2019_EurRespirJ

Zhu et al. Eur Respir J is now cited. However, we have not cited all previous asthma GWAS.

Reviewer #2:

This is an interesting and timely study providing an advancement in our understanding of asthma including the identification of genetic variants associated with disease risk but also importantly leveraging different approaches to identify the genes underpinning these observations. This provides new understanding but also new targets for drug development and repositioning. Overall the manuscript is divided into five main sections; i) GWAS of asthma in UKB, ii) TWAS using two lung tissue samples, iii) identification of eGenes in blood, iv) functional annotation of variants and v) evaluation of gene targets as drug opportunities. Overall, this approach works well with ultimately the authors highlighting 21 genes/proteins that warrant prioritisation in existing/new drug development for asthma. While an addition to the field several issues need resolving prior to potential publication.

Major comments

1. The GWAS in UKB using 56k cases and 352k controls is the largest to date mainly by leveraging some additional cases from records, could the authors clarify this and how this study compares to the now many studies using UKB for GWAS of asthma? In particular how different to that reported by Johansson et al 2019, PMID: 31361310. In the Johansson study 41k asthma cases were included with 239k controls. This study is not included in the current manuscript? In addition a quick look suggest that the Johansson study has previously reported several of the “novel” signals included in the current manuscript, e.g. ADAMTSL3.

The study by Johansson et al. is now fully integrated in the revised manuscript. We now have 6 instead of 8 novel loci: rs112119265-*CARD11* on chromosome 7 and rs4842921-*ADAMTSL3* on chromosome 15 were identified by Johansson et al.

We have modified Figure 2 (regional plots), the Results section as well as Supplementary Tables accordingly.

2. The authors report 73 loci with 116 independent signals in the GWAS and highlight nine potential novel signals (with the caveat outlined in 1). It is important to note that no replication is attempted for these potentially novel signals which is standard practice when reporting new genetic findings. While the same sample size is unlikely to be achieved, the authors should investigate these signals in additional cohorts to provide robustness even at nominal significance, e.g. 23&Me, GABRIEL cohorts.

This was also suggested by reviewer #1. See our response to comment #1.

3. In the GWAS, a statistical threshold of 5×10^{-8} is used which is historical when GWAS was conceived and prior to large scale imputation, can the authors justify this when now testing >35 million variants?

We added this as a limitation in the Discussion section, which also partly addresses comment #1 of reviewer #1:

For the asthma GWAS, we have tested 35,270,583 SNPs (MAF>0.0001) and used the conventional common-variant significance P value threshold of 5×10^{-8} . By using a more stringent threshold of 5×10^{-9} recommended for whole-genome sequencing studies including rare variants (PMID: 27990689), 18 out of 72 loci reported in our study would no longer be significant. Among them, 12 of these loci have been reported in previous asthma GWAS. All 18 loci have MAF>0.05 including the 6 novel loci with MAF ranging from 0.15 to 0.44. We did not observe replication of these novel loci in the Trans-National Asthma Genetic Consortium (Demenais et al. Nat Genet 2018, PMID 29273806) and they should thus be interpreted with caution.

4. The sensitivity analyses of the genetic association testing is well conceived to investigate other lung diseases, smoking and allergy. For any novel signals (see points 1 and 2 above) can the authors include a summary of the findings from the sensitivity analyses for the sentinel SNPs.

Summary of the findings from the sensitivity analyses for sentinel variants at all asthma-associated loci are now provided in Supplementary Table 5.

5. The TWAS approach using to sets of lung samples is a strength and a limitation, more details/discussion should be included regarding the subjects that donated material and the

potential heterogeneity with the lung eQTL dataset, e.g. cell composition/disease status (how many donors of lung samples had asthma?, COPD?) etc.

The demographic and clinical characteristics of the 1,038 subjects involved in the lung eQTL dataset are now described in Supplementary Table 20.

We have used whole lung and blood tissues, which we listed as a limitation in the Discussion section: We used whole lung and blood tissues, which contain heterogeneous cell populations, limiting our ability to identify genes affecting asthma risk through gene regulation and epigenetic marks. Progress in single-cell transcriptomic is promising for future studies.

6. The identification of deleterious variants in LD with the signals is useful, in addition to the current focus on p-values, could the authors specifically comment on effect sizes for these variants particularly when some striking e.g. stop codon in FLG.

The beta values were added to Table 2. This was added in the Results section:

In terms of effect sizes, the absolute beta values for the nine variants listed in Table 2 range from 0.03 to 0.22 (corresponding to ORs from 1.03 to 1.25). The largest effect size was observed for variant rs61816761 causing a G to A substitution (c.16819G>A) that occurs in exon 3 of the *FLG* gene, resulting in a stop instead of an arginine in codon 501 (p.Arg501Ter). However, the effect sizes of these coding variants were within the range observed for the 72 sentinel asthma-associated variants with ORs from 1.03 to 1.33. Noticeably, the largest effect size among sentinel variant was also observed for an independent deleterious coding variant in the *FLG* gene (rs558269137, p.Ser761CysfsX36).

7. IF I understand correctly, 24/55 genes identified in the Lung eQTL TWAS could not be evaluated in the replication GTEx dataset as expression data was not available. This is a major limitation and should be highlighted/discussed.

It is not expression data that is not available, it is the gene expression prediction models that are not significant in GTEx lung.

In the Methods section, we indicated that significant models were obtained for 19,918 probe sets in the lung eQTL dataset and for 11,518 genes in GTEx lung.

This paragraph was added in the revised manuscript to make this point clearer for the readers, which also addresses concerns raised by reviewer #1 (see comment #8):

Replication in this study is challenging as we have used the largest asthma GWAS study and the largest lung eQTL study. Similarly powered replication sets are currently not available. For the lung TWAS genes, our best attempt to replicate the novel findings was to use the lung eQTL set from GTEx.

Unfortunately, 24 out of 55 genes identified using the lung eQTL study did not yield significant gene expression prediction models and thus could not be evaluated by TWAS. This is partly the results of a smaller sample size in GTEx lung (n=515 vs 1,038), but many other factors. It should be emphasized that a head-to-head comparison between two eQTL sets is not straightforward as two GWAS sets and this not just because of the nature of data, i.e. static for SNP compared to dynamic and cell/tissue dependent for gene expression. There are major differences between our lung eQTL study and GTEx lung including the human lung sampling (surgery vs. post-mortem) and the gene expression platforms (microarray vs. RNA-Seq). Tissue processing methods for freezing, storing, and thawing tissues as well as extracting RNA were also different. Other investigators have also highlighted extensive heterogeneity in gene expression for the lung transcriptome data in GTEx due to sampling location in

the lung and treatment related changes such a mechanical ventilation (PMID 27588449). Taken all together, GTEx lung is not the most suitable replication set, but our best option now. Despite all this, we were pleasantly surprised to replicate 23 TWAS genes in GTEx lung.

8. ENCODE/Roadmap Epigenomics Project data is used to help investigate the link between functional marks/variants and genes. This is useful, however these datasets have limited respiratory relevant cells and so this should be acknowledged as a limitation.

We added this as a limitation in the Discussion section:

Finally, we used regulatory and functional annotations derived from ENCODE and Roadmap Epigenomics data to find cell type and tissue enrichment of asthma-associated loci. Although valuable, these publicly available datasets lack functional data on some cell types that are relevant to asthma such as eosinophils and airway smooth muscle cells, which limited our ability to understand the functional impact of asthma-associated variants.

9. For the blood based analyses involving 31,684 samples, why was a discovery/replication approach not used to provide confidence in findings?

As indicated for point #9 of reviewer #1, the blood eQTL dataset used in this study is from the eQTLGen Consortium (Võsa et al. biorxiv 2018) and we only have access to summary statistics.

10. The current manuscript would greatly benefit from a direct comparison of the main findings of the current study and discussion of a similar paper that used published GWAS data, eQTL datasets and drug information to identify priority candidates for drug development in asthma see PMID: 32910899.

We read with great interest the manuscript by El-Husseini et al. that provided a comprehensive list of asthma drug targets derived from GWAS and eQTL studies.

We have cross-referenced the list of 161 possible asthma drug targets derived from El-Husseini et al. with genes prioritized in our manuscript. These changes were made:

-We added a column in Supplementary Table 17 that shows genes that overlap between the 806 prioritized genes identified in our study and the 161 asthma drug targets from El-Husseini et al.

-We also added a column in Table 6 to show this overlap.

-Finally, the Results section was modified:

Together, 806 unique target genes were identified with overlap across methods shown in Fig. 7.

Notably, 101 of them overlapped with the recent list of 161 possible asthma drug targets summarized by El-Husseini et al. (Lancet Respir Med 2020), which were derived from eQTL and non-synonymous analysis of independent variants associated with asthma in previous GWAS (Supplementary Table 17).

Minor comments

1. Throughout “top SNPs” are referred to, it is unclear how this is defined.

We have replaced the terminology “top SNP” and “top gene” by sentinel SNP/variant or most significant SNP/gene, as appropriate.

2. Table 3 would benefit from being focussed to replicated findings in both lung samples with all data put in Suppl.

Reviewer #1 (minor comment #8) has proposed changes in Table 3 that we believe address this concern.

Reviewer #3:

In this manuscript, the authors leveraged lung and blood transcriptome as well as epigenetic marks to prioritize the most likely causal genes within asthma susceptibility loci derived from the UK Biobank. The authors should be commended in their use of various bioinformatics approaches. I have a couple of comments:

1. There is no functional validation to investigate the biological role of the candidate causal genes in the pathogenesis of asthma. Future follow-up studies are needed to demonstrate causality in order to address the clinical needs of asthma patients. It would be good if the authors could provide more evidence on biological relevance and clinical implications of the prioritized candidate causal genes and the added value of your findings.

We agree that follow-up studies are needed. It is a major challenge to address the variant-to-gene-to-function in a single manuscript.

We indicated this limitation in the Discussion:

The candidate gene targets identified in our study will need to be experimentally validated by other preclinical models (cellular, animal and human studies) in order to understand the biological effects of risk alleles on gene function and the role of these genes in the pathogenesis of asthma. Further studies are needed to demonstrate causality.

2. Asthma is a clinically heterogeneous disease, and distinct loci have been reported underlying the susceptibility to childhood-onset asthma and adult-onset asthma (PMID: 31036433). Previous gene expression and tissue enrichment patterns suggest that genes at the childhood onset loci were most highly expressed in skin, blood and small intestine, whereas genes at the adult onset loci were most highly expressed in lung, blood, small intestine and spleen. Therefore, the role of age of asthma onset should be taken into account in GWAS to prioritize candidate causal genes, and TWAS analyses should also be conducted in the other asthma relevant tissue such as skin, small intestine and spleen that are available in GTEx consortium.

There are two previous studies that have used UK Biobank to study the genetics of childhood vs. adult-onset asthma (PMID: 31036433 & 30929738). In our study, we decided to focus on a broad asthma definition. Post-GWAS analyses by subgroups (childhood vs. adult-onset, male vs. female, etc.) is a different manuscript.

We have now performed TWAS analysis using skin, small intestine and spleen from GTEx. We have also included the TWAS in blood based on a comment raised by reviewer #1 (see comment #9).

This was added in the Methods section:

TWAS in GTEx blood, skin, small intestine and spleen

S-PrediXcan was also used to explore TWAS genes in other asthma relevant tissues. GTEx v8 datasets for blood (n=670), skin not sun exposed (n=517), skin sun exposed (n=605), small intestine (n=174)

and spleen (n=227) were analysed with the asthma GWAS. Significant gene expression models (FDR<0.05) were obtained for 10,210 genes in blood, 12,347 genes in skin not sun exposed, 13,375 genes in skin sun exposed, 5,184 genes in small intestine, and 8,473 genes in spleen. Bonferroni-corrected TWAS genes were thus set at $P_{\text{TWAS}} < 4.90\text{E-}6$, $P_{\text{TWAS}} < 4.05\text{E-}6$, $P_{\text{TWAS}} < 3.74\text{E-}6$, $P_{\text{TWAS}} < 9.65\text{E-}6$, and $P_{\text{TWAS}} < 5.90\text{E-}6$, respectively.

This was added in the Results section:

Asthma TWAS in other asthma-relevant tissues in GTEx

Asthma TWAS genes were also explored in five additional tissues in GTEx, namely blood, skin (exposed or not to sun), small intestine and spleen. Genome-wide TWAS results for all tissues are illustrated in Fig. 4. The numbers of TWAS genes reaching significance were 63 in blood, 65 for skin not sun exposed, 66 for skin sun exposed, 27 for small intestine, and 34 for spleen. Interestingly, many of these genes overlapped with those identified using lung data (Fig. 4). We have also evaluated overlapping TWAS genes across GTEx tissues (Supplementary Fig. 7). Genes identified in at least 4 out of 5 tissues include those on 2q37.3 (*D2HGDH*), 5q31.1 (*SLC22A5*, *KIF3A*), 6p22-p21-MHC (*HLA-DQA1*, *HLA-DQA2*, *HLA-DQB1*, *HLA-DQB1-AS1*, *HLA-DRB1*), 6q23.3 (*AH11*), 12q13.2 (*RPS26*, *SUOX*), and 17q12-q21.2 (*ORMDL3*, *GSDMA*, *GSDMB*, *PGAP3*, *MED24*). Among them, *SLC22A5*, *KIF3A*, *HLA-DQB1*, *ORMDL3*, *GSDMA*, and *GSDMB* were also identified in lung. The full summary statistics for the asthma TWAS using blood, skin (sun exposed or not), small intestine and spleen are available in the Supplementary Data 2 to 6.

Figure 4. Manhattan plots of the TWAS on asthma integrating the UK Biobank GWAS and the eQTL from five tissues in GTEx. From top to bottom, TWAS results for blood, skin not sun exposed, skin sun exposed, small intestine and spleen are illustrated. Each dot represents the association between predicted gene expression and asthma for a specific gene/transcript. P values for gene expression-asthma associations are on the y axis in $-\log_{10}$ scale. The green and magenta horizontal lines represent P_{TWAS} of 0.0001 and Bonferroni, respectively. Annotations for genome-wide significant genes/transcripts that passed Bonferroni correction are indicated. Genes in blue have also been identified in lung tissue.

Supplementary Figure 7. Upset plot showing overlap of TWAS genes across GTEx tissues. TWAS genes found in four and five tissues are annotated. Lung TWAS genes are in bold.

3. What is the age range of samples in the eQTL datasets used in this analysis? If the samples were mainly from adults, this study may miss relevant genes whose expression is environmentally regulated.

Based on comment 5 of reviewer #2, we added demographic and clinical characteristics of the 1,038 subjects that participated in the lung eQTL dataset (see Supplementary Table 20). As indicated in the text, we obtained our lung specimens from patients that underwent lung surgery.

We made modifications in the Discussion:

Environmental risk factors and the specific period of exposures during the lifespan play an important role in the development of asthma. Our genomic datasets (GWAS and eQTL) are retrospective in nature and mostly derived from adult populations. Environmental and age-related modifiers of genetic risk and gene expression levels are likely to have been missed.

4. The authors did not provide any replication of the asthma loci identified in the UK Biobank. The lack of a replication data set should be listed as a limitation.

This was also suggested by the other reviewers. See our response to comment #1 of reviewer #1.

5. GARFIELD analysis showed strong enrichment in a variety of cell types in blood including GM12878 cell line, CD19_Primary_Cells, CD4_Primary_Cells and CD8_Primary_Cells, it would be helpful to additionally use Hi-C data obtained in other cell types in blood for chromatin contact mappings.

The top cell line showing enrichment in GARFIELD is GM12878, which is a lymphoblastoid B cell line. Hi-C data for GM12878 is publicly available:

<https://www.ncbi.nlm.nih.gov/geo/query/acc.cgi?acc=GSE87112>

Adding Hi-C data from other cell lines would be feasible, but we start to feel that there are a lot of data in the revised manuscript (7 Figures, 6 Tables, 8 Suppl. Figures, and 20 Supplementary Tables). So we would prefer to keep the focus on the cell line showing the most enrichment in GARFIELD.

6. What is the possible reason that lung is not significant for enrichment of DNase I hypersensitive sites?

We were surprised by this result and we can only speculate on possible reasons. Reviewer #2 (see comment 8) has pointed out a limitation of our study that may explain, at least partly, why the lung is not significantly enriched in DHS.

This was added as a limitation in the Discussion:

We used whole lung and blood tissues, which contain heterogeneous cell populations, limiting our ability to identify genes affecting asthma risk through gene regulation and epigenetic marks. Progress in single-cell transcriptomic is promising for future studies. Finally, we used regulatory and functional annotations derived from ENCODE and Roadmap Epigenomics data to find cell type and tissue enrichment of asthma-associated loci. Although valuable, these publicly available datasets lack functional data on some cell types that are relevant to asthma such as eosinophils and airway smooth muscle cells, which limited our ability to understand the functional impact of asthma-associated variants.

7. It has been suggested that, for validating drug targets, the gene must include multiple causative variants of known biological function (PMID: 23868113). As some asthma loci identified in this study contain independent variants that are not in LD, these SNPs could be used to investigate dose–response curves. It would be more convincing evidence if the genes prioritized as therapeutic targets for asthma could demonstrate a dose–response relationship.

Thank you for suggesting the read of Plenge et al. To use the same terminologies as Plenge et al., we have certainly not provided a complete package to support novel therapeutic hypotheses, in comparison, for example, of LDL cholesterol (and naturally occurring variants in *LDLR*, *PCSK9* and *HMGCR* genes) and heart disease. The genetic architecture of Mendelian vs. complex diseases is also

well explained in Plenge et al. (see Box 1). Asthma is a complex disease with no monogenic form of the disease identified yet. We are thus expecting to find disease-associated alleles with more subtle effects on the phenotype. As expected from GWAS on complex diseases, most associated variants fall outside of protein-coding sequences and are more likely regulate gene expression (directly or by LD). In this study, we took advantage of eQTL studies to test whether asthma alleles raise or lower gene expression levels (in disease relevant tissues) and subsequently influence the risk of asthma. The TWAS approach captures the effect of independent variants on gene expression and then assesses the association between the cis-genetic component of expression and asthma. Similarly, for two-sample MR analysis, the instruments are built from independent variants associated with gene expression. Then, the aggregate effect of these variants is tested on the outcome (here asthma) and used to differentiate between cause and consequence. TWAS and MR are thus similar concepts to the dose-response curves from allelic series (multiple independent alleles) proposed by Plenge et al. However, in our study, we used an intermediate phenotype (i.e. gene expression) to uncover the small effects of disease-associated alleles derived from a GWAS on a complex disease.

Plenge et al. have stressed “that human genetics represents the first step towards a complete package for drug development”. We believe that our study checked that first box and goes beyond by showing the direction of effect of asthma risk variants on gene expression levels in disease-relevant tissues and by screening for druggable gene targets. However, more studies will be needed to build a complete package for drug development.

This section was modified in the Discussion:

We used the best possible bioinformatics approaches to identify causality genes. However, our study does not provide a complete package to support novel therapeutic hypotheses (Plenge et al.). The candidate gene targets identified in our study will need to be experimentally validated by other preclinical models...

REVIEWERS' COMMENTS:

Reviewer #3 (Remarks to the Author):

The authors have addressed my comments satisfactorily. Please note that "Regional plots showing the six (rather than nine) new asthma-associated loci" in the legend of Fig.2.

Responses to reviewers

We would like to thank the reviewers for taking the time to evaluate our revised manuscript. You will find below a point-by-point response to comments. The comments from the reviewers are provided verbatim in bold.

Reviewer #1:

1. I am happy with the text added regarding replication. When I looked at the full TAGC summary statistics available from GWAS catalog I found rs9479812, reported with $P = 0.02$, is in LD with the reported signal rs7770794 ($r^2=0.904$ according to LDlink European), the authors may want to look at this?

The result for our sentinel variant at that locus (rs7770794) is provided in the summary statistics from TAGC (P value = 0.54). As the reviewer know, one SNP in LD with $P = 0.02$ is not convincing to claim replication. We have thus generated the regional plot for this locus (chr6:154127823-155128603) using summary statistics from TAGC and forcing rs7770794 as the sentinel variant. We found no SNP in LD with rs7770794 with P value < 0.05 . We have thus kept our conclusion that this locus is not replicated in TAGC.

2. I'd emphasise the new signals being reported e.g. prepend the new text:

“We report 4 novel signals, two independent signals at the MHC locus (rs2517761 and rs2523430) and two at the 1q21-FLG locus (rs185433896 and rs558312428) were independent ($r^2 < 0.1$) from asthma-associated variants reported in the literature (Supplementary Table 3).”

We have changed as suggested.

3. Happy with supplementary table 1.

In Supplementary Table 4 I don't understand why there are “Asthma GWAS in UK Biobank from this study” variants listed with P going down to 1.49E-47 if this is meant to be previous results that are *not* significant in this study?

Thanks for noticing this mistake. Variants that were significant ($P < 5.0E-8$) in ST4 were from Ferreira_2017_NatGenet and they should not be in this table. The error originates from the fact that the chromosome number was missing (column “Chr”) for these variants. The correction was made and now no variant in ST4 (SD4 in the revised manuscript) is significant in our study.

7. I'd perhaps use more positive tone, say “Among the 55 lung TWAS genes, nine are novel for asthma” and label them as novel in the figure and table rather than “not previously reported”, but no strong opinion.

We have changed as suggested.

Reviewer #3:

The authors have addressed my comments satisfactorily. Please note that “Regional plots showing the six (rather than nine) new asthma-associated loci” in the legend of Fig.2.

Done. Thank you.